# LIKELIHOOD TRAINING OF CASCADED DIFFUSION MODELS VIA HIERARCHICAL VOLUME-PRESERVING MAPS

**Henry Li**[1]**, Ronen Basri**[2,3]**, Yuval Kluger**[1]
[1]Yale University, [2]Meta AI, [3]Weizmann Institute of Science
`{henry.li, yuval.kluger}@yale.edu`
`ronen.basri@weizmann.ac.il`

## ABSTRACT

Cascaded models are multi-scale generative models with a marked capacity for producing perceptually impressive samples at high resolutions. In this work, we show that they can also be excellent likelihood models, so long as we overcome a fundamental difficulty with probabilistic multi-scale models: the intractability of the likelihood function. Chiefly, in cascaded models each intermediary scale introduces extraneous variables that cannot be tractably marginalized out for likelihood evaluation. This issue vanishes by modeling the diffusion process on latent spaces induced by a class of transformations we call hierarchical volume-preserving maps, which decompose spatially structured data in a hierarchical fashion without introducing local distortions in the latent space. We demonstrate that two such maps are well-known in the literature for multiscale modeling: Laplacian pyramids and wavelet transforms. Not only do such reparameterizations allow the likelihood function to be directly expressed as a joint likelihood over the scales, we show that the Laplacian pyramid and wavelet transform also produces significant improvements to the state-of-the-art on a selection of benchmarks in likelihood modeling, including density estimation, lossless compression, and out-of-distribution detection. Investigating the theoretical basis of our empirical gains we uncover deep connections to score matching under the Earth Mover's Distance (EMD), which is a well-known surrogate for perceptual similarity. Code can be found at this https url.

## 1 INTRODUCTION

A widely used strategy for generating high resolution images $\mathbf{x}$ is to first draw from a low resolution model $p_\theta(\mathbf{z}^{(1)})$, then refine this representation with a series of super-resolution models $p_\theta(\mathbf{z}^{(s)}|\mathbf{z}^{(s-1)})$ for $k = 2, \ldots, S$ where $\mathbf{z}^{(S)} = \mathbf{x}$. This idea, known as cascaded or multi-scale modeling, has been employed to great effect in generative models at large (Oord et al., 2016; Karras et al., 2017; De Fauw et al., 2019), as well as diffusion models in particular (Ho et al., 2022). However, multi-scale models pose a fundamental issue for likelihood computation, as the modeled likelihood function — obtained as a product of all conditional super-resolution models and the base model — is now the joint likelihood $p_\theta(\mathbf{z}^{(1)}, \ldots, \mathbf{z}^{(K)})$ rather than the desired likelihood $p_\theta(\mathbf{x}) = p_\theta(\mathbf{z}^{(K)})$. This prevents cascaded models from being used for likelihood-based training and inference on $\mathbf{x}$. For $p_\theta(\mathbf{x})$ to regain exact evaluation capabilities, the intermediate resolutions must must either be marginalized out — a very expensive operation in high dimensions — or undergo complex redesign to avoid such marginalization (Menick & Kalchbrenner, 2018; Reed et al., 2017).

In this work, we propose a different approach to overcoming this problem using a class of transformations we call hierarchical volume-preserving maps. These are a special subset of homeomorphisms to which the likelihood function is invariant — i.e., a class of functions $\mathcal{H}$ such that for all $h \in \mathcal{H}$, $p_\theta(\mathbf{x}) = p_\theta(h(\mathbf{x}))$. Under these transformations, the joint likelihood computed by a cascading model coincides directly with the desired likelihood function $p_\theta(\mathbf{x})$. When these transformations are simple and produce useful *hierarchical* representations of $\mathbf{x}$, we are able to retain maximum likelihood

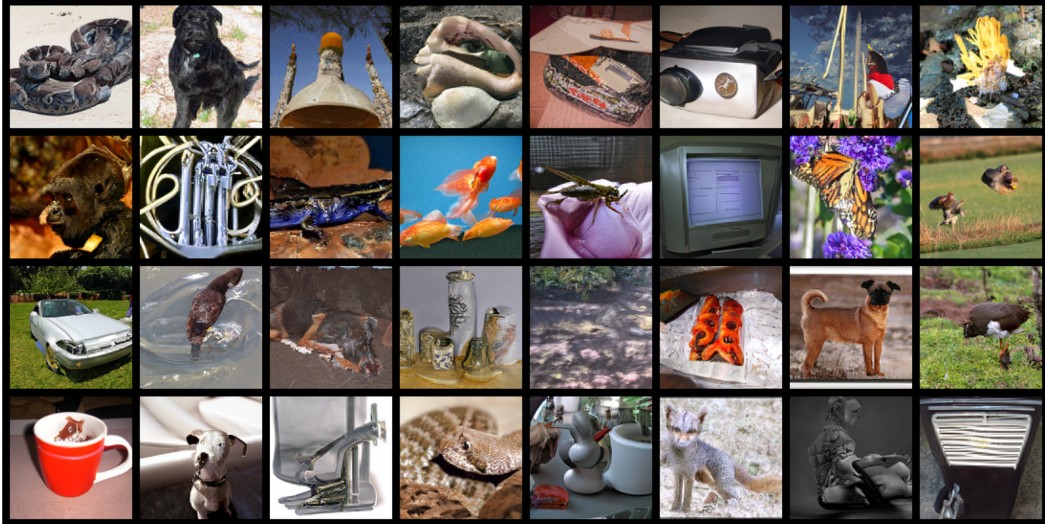

Figure 1: Images generated from our W-PCDM model trained on unconditional ImageNet 128x128.

training and evaluation capabilities in a cascaded multi-scale setting while introducing little to no computational overhead. We evaluate our multi-scale likelihood model on a selection of datasets and tasks including density estimation, lossless compression, and out-of-distribution detection and observe significant improvements to the existing state-of-the-art, demonstrating the power behind a multi-scale prior for likelihood modeling.

We subsequently turn to a theoretical analysis of our framework, and show that maximizing the log-likelihood of our proposed model is equivalent to minimizing an upper bound on the weighted sum (or integral) of Earth Mover's Distances (EMD) on the marginal scores in the diffusion process. The Earth Mover's — or alternatively Wasserstein-$1$ — Distance is a commonly used metric in computer vision known for its properties as a high quality surrogate for human perceptual similarity between images (Rubner et al., 2000). However, training a diffusion model using this metric between images is expensive and grows as $\mathcal{O}(N^3 \log N)$ with the image size. Surprisingly, by utilizing a multi-scale basis, our upper bound can be evaluated in $\mathcal{O}(N)$ time, which costs the same as a standard $L^2$ norm on images. This constitutes a dramatic speed-up that greatly increases the feasibility of training with EMD. We theorize that this special property underpins the empirical success of our model in density estimation and downstream tasks.

Our contributions can be summarized as follows:

- We demonstrate the existence of a class of transformations we call *hierarchical volume-preserving maps* to which the likelihood function is invariant. Leveraging this class of mappings, we construct a multi-scale cascaded diffusion model that recovers the likelihood training characteristics of standard diffusion models without requiring an expensive marginalization step.

- We consider that two well-known multiscale decompositions, the Laplacian pyramid and wavelet transforms, and show that they are hierarchical volume preserving maps, where the linearity of the maps reduce the volume-preserving property into more familiar notions of orthogonality and tight frames (Unser et al., 2011). We establish the capabilities of our model with significant improvements in likelihood modeling, lossless compression, and out-of-distribution detection performance on a selection of standard image-based density estimation benchmark datasets.

- Investigating the theoretical basis for this performance gain, we uncover a deep theoretical connection between EMD score matching and cascaded diffusion models trained via two special classes of hierarchical volume-preserving maps, the wavelet and Laplacian pyramid maps.

## 2 RELATED WORK

**Likelihood Training**    The maximum likelihood training of generative models involves designing a model with a tractable likelihood $p_\theta(\mathbf{x})$ — i.e., the probability that a model with parameters $\theta$ generates an image $\mathbf{x}$ — and maximizing this function with respect to $\theta$. This practice can be traced back to the wake-sleep algorithm (Hinton et al., 1995), which is used to select parameters for deep belief networks and Helmholtz machines (Dayan et al., 1995). More modern treatments include variational autoencoders (VAEs) (Kingma & Welling, 2013), normalizing flows (NFs) (Dinh et al., 2014), and autoregressive models (ARMs) (Van Den Oord et al., 2016; Salimans et al., 2017). Diffusion models also fit into this category, as they were originally formulated in a variational Bayesian framework (Sohl-Dickstein et al., 2015). However, they are often trained with a non-probabilistic loss function (Ho et al., 2020; Song et al., 2020b) due to the instability of the originally derived loss. Recently, (Song et al., 2021) explore likelihood training in a frequentist setting, whereas (Chen et al., 2021) do so via the lens of the Schrodinger Bridge problem and Forward-Backward SDEs, while (Kingma et al., 2021) introduce several modifications to the originally derived probabilistic loss function that greatly improve its stability during training and inference. But there is to our knowledge no known likelihood-based treatment of diffusion models in a multi-scale setting, to which we turn our attention.

**Multiscale Generative Models**    Hierarchical modeling of spatially structured data with progressively growing resolution scales has seen widespread use across image (Karras et al., 2017; Child, 2020; Han et al., 2022) and audio (Oord et al., 2016) domains. A hierarchical structure encourages the data synthesis task to be split into smaller sequential steps, allowing models to learn spatial correlations at each resolution scale separately, and counteracts the tendency to overly focus on local structure (De Fauw et al., 2019). In normalizing flows, (Yu et al., 2020) construct a multiscale flow by leveraging the orthogonality of the wavelet transform, which can be seen as a volume-preservation condition on linear equidimensional transforms — our Eq. 8 is a generalization of this observation. In diffusion models, (Ho et al., 2022); (Guth et al., 2022; Wang et al., 2022) formulate a non-probabilistic hierarchical framework for diffusion models and demonstrate its effectiveness in generating high resolution images at impressive levels of perceptual fidelity. In spite of these advances, likelihood training of multi-scale models is generally unwieldy, requiring expensive marginalization, loose variational bounding, or complex factorization schemes where pixels in low-resolution images must be directly utilized as pixels in the higher resolution images (Reed et al., 2017; Menick & Kalchbrenner, 2018).

**Diffusion Modeling with Earth Mover's Distances**    We also show that likelihood training with our hierarchical model approximates score matching with an optimal transport cost. Existing works have also noted relationships between diffusion modeling and optimal transport (De Bortoli et al., 2021; Chen et al., 2021; Lipman et al., 2022; Shi et al., 2022; Kwon et al., 2022), showing that diffusion models learn to generate data by transporting samples $\mathbf{x} \in \mathbb{R}^d$ along paths that minimize a cost between the prior $p(\mathbf{x}_T) \sim \mathcal{N}(\mathbf{0}, \mathbf{I})$ and data distribution $p(\mathbf{x}_0)$. Our result is markedly different: we show that our model learns the solution to an optimal transport problem in the two-dimensional coordinate space of pixel histograms. That is, we consider the pixel values of each image as an unnormalized and discretized histogram on a square in $\mathbb{R}^2$, and optimize a transport cost over this grid. This metric on images (as well as other spatial data) is well-studied, and known to correlate well with perceptual similarity (Rubner et al., 2000; Zhang et al., 2020; Zhao et al., 2019). Our perpespective is thus orthogonal and complementary to prior work.

## 3 BACKGROUND

Diffusion models (Ho et al., 2020; Song et al., 2020b) are inspired by non-equilibrium thermodynamics (Sohl-Dickstein et al., 2015) designed to model $p_\theta(\mathbf{x}) = \int p_\theta(\mathbf{x}_{0:T}) \, d\mathbf{x}_{1:T}$ where data $\mathbf{x}_0 := \mathbf{x}$ are related to a set of latent variables $\mathbf{x}_{1:T}$ by a diffusion process. Namely, $\mathbf{x}_{1:T} := (\mathbf{x}(t_1), \ldots, \mathbf{x}(t_T))$ are distributed as marginals of a process governed by an Itô stochastic differential equation (SDE)

$$d\mathbf{x} = \mathbf{f}(\mathbf{x}, t) \, dt + g(t) \, d\mathbf{w} \qquad (1)$$

evaluated at times $\{t_k\}_{k=1}^T$. $\mathbf{f}$ and $g$ are typically called *drift* and *diffusion* functions, and $\mathbf{w}$ is the standard Wiener process. Samples can then be generated by modeling the reverse diffusion, which

has a simple form given by (Anderson, 1982)

$$d\mathbf{x} = [\mathbf{f}(\mathbf{x}, t) - g(t)^2 \underbrace{\nabla_{\mathbf{x}} \log q(\mathbf{x}, t|\mathbf{x}_0)}_{\approx \mathbf{s}_\theta(\mathbf{x}, t)}] dt + g(t) d\overline{\mathbf{w}}, \tag{2}$$

where $\overline{\mathbf{w}}$ is the corresponding reverse-time Wiener process. Training the diffusion model involves approximating the true score function $\nabla_{\mathbf{x}} \log q(\mathbf{x}, t|\mathbf{x}_0)$ with a neural network $\mathbf{s}_\theta(\mathbf{x}, t)$ in Eq. (2). This can be achieved directly via score matching (Hyvärinen & Dayan, 2005; Song & Ermon, 2019; Song et al., 2020b), or by modeling the sampling process (Sohl-Dickstein et al., 2015; Ho et al., 2020; Kingma et al., 2021), which is obtained by approximating the marginal distributions of the discretized reverse-time Markov chain $q$ with a parametric model $p_\theta$, i.e.,

$$q(\mathbf{x}_{1:T}|\mathbf{x}_0) = q(\mathbf{x}_T) \prod_{k=1}^{T-1} q(\mathbf{x}_k|\mathbf{x}_{k+1}, \mathbf{x}_0) \quad (3) \quad \text{and} \quad p_\theta(\mathbf{x}_{0:T}) = p(\mathbf{x}_T) \prod_{k=0}^{T-1} p_\theta(\mathbf{x}_k|\mathbf{x}_{k+1}). \tag{4}$$

Maximum likelihood estimation can then be performed by optimizing a tractable likelihood bound $\log p_\theta(\mathbf{x}_0) \geq \mathcal{L}(\mathbf{x}) =$

$$\mathbb{E}_{\mathbf{x}_{1:T} \sim q}\left[ \log p_\theta(\mathbf{x}_0|\mathbf{x}_1) - KL(q(\mathbf{x}_T|\mathbf{x}_0)||p_\theta(\mathbf{x}_T)) - \sum_{k=1}^{T-1} \log KL(q(\mathbf{x}_k|\mathbf{x}_{k+1}, \mathbf{x}_0)||p_\theta(\mathbf{x}_k|\mathbf{x}_{k+1})) \right], \tag{5}$$

which forms the basis for variational inference with diffusion models. Several further improvements were proposed by (Kingma et al., 2021), such as antithetic time sampling, a continuous-time relaxation of the Markov chain, and a learnable noise schedule parameterized by a monotonic neural network. Finally, unbiased estimates of $\log p_\theta(\mathbf{x})$ can be obtained by modeling the probability flow ODE (PF-ODE) corresponding to Eq. (2) from (Song et al., 2020b)

$$d\mathbf{x} = \left[ \mathbf{f}(\mathbf{x}, t) - \frac{1}{2} g(t)^2 \nabla_{\mathbf{x}} \log p(\mathbf{x}, t) \right] dt, \tag{6}$$

using the continuous change of variables formula (Chen et al., 2018) and substituting the score with the score network $\mathbf{s}_\theta(\mathbf{x}, t)$.

To build a *cascaded* diffusion model, $\mathbf{x}$ is further decomposed into a series of latent representations $\{\mathbf{z}^{(s)}\}_{s=1}^S$ of decreasing scales, and $p_\theta(\mathbf{z}^{(s)}|\mathbf{z}^{(s-1)})$ is approximated. Now, the data may be sampled autoregressively — drawing first from a low-resolution prior $\mathbf{z}^{(1)} \sim p_\theta(\mathbf{z}^{(1)})$, then from a series of conditional super-resolution models $\mathbf{z}^{(s)} \sim p_\theta(\mathbf{z}^{(s)}|\mathbf{z}^{(s-1)})$ for $s = 2, \ldots, S$ and where $\mathbf{z}^{(S)} = \mathbf{x}$. However, it is in this hierarchical setting that we run into the aforementioned intractability of the likelihood function, where $p_\theta(\mathbf{z}^{(S)} = \mathbf{x})$ can only be recovered by marginalizing the modeled likelihood over each latent scale, i.e.,

$$p_\theta(\mathbf{x}) = \int_{\mathbf{z}^{(1:S-1)}} p_\theta(\mathbf{z}^{(1)}, \mathbf{z}^{(2)}, \ldots, \mathbf{z}^{(S)}) d\mathbf{z}^{(1)} d\mathbf{z}^{(2)} \ldots \mathbf{z}^{(S-1)}. \tag{7}$$

## 4    HIERARCHICAL VOLUME-PRESERVING MAPS

Due to the intractability of the likelihood function in many hierarchical models, we are interested in multi-scale decompositions of $\mathbf{x}$ that can sidestep the need for marginalization during likelihood computation. In this section, we shall demonstrate the existence of a subset of homeomorphisms $\mathcal{H}$ we call hierarchical volume-preserving maps to which the likelihood function $p_\theta$ displays a form of probabilistic invariance — i.e., $p_\theta(\mathbf{x}) = p_\theta(h(\mathbf{x}))$ for all $h \in \mathcal{H}$. The utility of such transformations is clear: if $h(\mathbf{x})$ is the scales of the hierarchical model, then we can directly use the joint likelihood over these scales as the desired likelihood function. Finally, we consider two simple homeomorphisms that satisfy our proposed condition: the Laplacian pyramid and wavelet transforms.

## 4.1 A Probabilistic Invariance

To build towards this result, we review the notion of *volume-preserving maps*, which can be understood as transformations that are constrained with respect to a local measure of space expansion or contraction, the matrix volume.

**Definition 1** (Volume-preserving Maps (Berger & Gostiaux, 2012))**.** *A homeomorphic (i.e., smooth, invertible) transformation $h : \mathcal{X} \to \mathcal{Z}$ is known to be volume-preserving if it everwhere satisfies*

$$\sqrt{det\left(A^T A\right)} = 1, \tag{8}$$

*where $det(\cdot)$ is the determinant and $A \in \mathbb{R}^{dim(\mathcal{Z}) \times dim(\mathcal{X})}$ is the Jacobian of $h(\mathbf{x})$.*

Crucially, $\dim(\mathcal{Z})$ may be larger than $\dim(\mathcal{X})$ — $h$ may be a manifold-valued function, so long as $A$ is full rank (i.e. $\text{rank}(A) \geq \dim(\mathcal{Z})$). Our notion of *hierarchical* volume-preserving maps further requires that the transformation is multi-scale (Mallat, 1999; Burt & Adelson, 1987; Horstemeyer, 2010), i.e., $h : \mathcal{X} \to \mathcal{Z}^{(1)} \times \cdots \times \mathcal{Z}^{(S)}$, where the original data $\mathbf{x} \in \mathcal{X}$ is decomposed into a sequence of representations $\mathbf{z}^{(s)} \in \mathcal{Z}^{(s)}$ that describe the data at varying scales — an essential property for cascaded modeling. The following result reveals the utility of working with such transformations.

**Lemma 4.1** (Probabilistic Invariance of Hierarchical Volume-preserving Maps)**.** *Let $h$ be a hierarchical volume-preserving map such that $h(\mathbf{x}) = (\mathbf{z}^{(1)}, \mathbf{z}^{(2)}, \ldots, \mathbf{z}^{(S)})$, and $p_\theta$ be a likelihood function on $\mathbf{z}^{(1)}, \mathbf{z}^{(2)}, \ldots, \mathbf{z}^{(S)}$. Then the likelihood function with respect to the original data $p_\theta(\mathbf{x})$ can be recovered by the simple relation*

$$\log p_\theta(\mathbf{x}) = \log p_\theta[h(\mathbf{x})]. \tag{9}$$

In other words, the joint likelihood given by the cascaded model under a hierarchical volume-preserving map is precisely the desired likelihood function with respect to the data $p_\theta(\mathbf{x})$, and we no longer require any additional computation or design considerations to remove the intermediary scales. In this sense, Eq. 9 displays a probabilistic invariance to $h$ [1].

**Standard Cascaded Hierarchy**    We now demonstrate how the standard hierarchy of a cascaded model such as in (Karras et al., 2017) or (Ho et al., 2022) is not a hierarchical volume-preserving map. Here, the scales $\mathbf{z}^{(1)}, \mathbf{z}^{(2)}, \ldots, \mathbf{z}^{(S)}$ are subsampled versions of the full-resolution sample $\mathbf{x}$, with each scale twice the resolution of the preceding scale. Consider a simple illustrative example in $\mathbb{R}^d$. When using the nearest neighbor resampling method, $h$ may be defined as the concatenation of $S$ different identity-like functions

$$h = [\mathbf{I}_{d \times d}, 2^{-1} \cdot \mathbf{I}_{d_1 \times d_1}^{(2)}, 2^{-2} \cdot \mathbf{I}_{d_2 \times d_2}^{(3)}, \ldots, 2^{-(S-1)} \cdot \mathbf{I}_{d_{S-1} \times d_{S-1}}^{(S)}], \tag{10}$$

where $d_k := d/2^k$ and $\mathbf{I}_{d \times d}^\ell$ denotes a $d \times d$ identity matrix where each column is repeated $\ell$ times. Inspecting this operator we find that its determinant grows exponentially with $d$. Therefore, while hierarchical, the standard transformation is certainly not volume-preserving. Accordingly, as has been established, there is no tractable form for the likelihood function $p_\theta(\mathbf{x})$ in terms of the joint likelihood $p_\theta(\mathbf{z}^{(1)}, \mathbf{z}^{(2)}, \ldots, \mathbf{z}^{(S)})$.

## 4.2 Special Instances of Hierarchical Volume-preserving Maps

While our formulation is fully general for arbitrary homeomorphisms $h$ that satisfy Eq. 8, we highlight two special and well-known classes of maps that are particularly noteworthy for their ease of implementation, computational tractability, and theoretical properties (Section 5.2).

**Laplacian Pyramids**    Though the standard cascaded hierarchy is not a hierarchical volume-preserving map, it is closely related to the Laplacian pyramid. This representation involves a set of band-pass images, each spaced an octave apart, obtained by taking differences between a sequence of low-pass filtered images with their unfiltered counterparts. Formally, consider base images

---

[1]Formally, letting $g_\mathbf{x}(h) = p_\theta(h(\mathbf{x}))$, the set of hierarchical volume-preserving maps $\mathcal{H}$ form an equivalence class under the relation $g_\mathbf{x}(h) = g_\mathbf{x}(h')$ for $h, h' \in \mathcal{H}$ and $\mathbf{x} \in \mathcal{X}$.

$\mathbf{x}$ of size $j \times j$ and let $d : \mathcal{R}^{j \times j} \to \mathcal{R}^{(j/2) \times (j/2)}$ and $u : \mathcal{R}^{j \times j} \to \mathcal{R}^{(j \cdot 2) \times (j \cdot 2)}$ denote downsampling and upsampling operators respectively. In this work, we choose them to be norm-preserving bilinear resamplers (i.e., images are scaled by a factor of 2 every downsample and a factor of $1/2$ every upsample). Then $\mathbf{z}^{(S)}$ can be written as

$$\mathbf{z}^{(S)} = \mathbf{x} - u(d(\mathbf{x})), \tag{11}$$

and the intermediary multi-scale representations $\mathbf{z}^{(2)}, \mathbf{z}^{(2)}, \dots, \mathbf{z}^{(S-1)}$ may be constructed via the recursive relation

$$\mathbf{y}^{(s)} = d(\mathbf{y}^{(s+1)}), \quad \text{and} \quad \mathbf{z}^{(s)} = \mathbf{y}^{(s)} - u(d(\mathbf{y}^{(s)})) \tag{12}$$

for $s = 2, \dots, S - 1$ and $\mathbf{y}^{(S)} := \mathbf{x}$. Finally, we define the base scale containing the low-frequency residual as $\mathbf{z}^{(1)} = \mathbf{y}^{(1)} = d(\mathbf{y}^{(2)})$.

We now verify that the Laplacian pyramid is a hierarchical volume-preserving map. Clearly, the Laplacian pyramid is multi-scale given a proper choice of $d$ and $u$. In our case, we simply choose the bilinear resampler. Moreover, the Laplacian pyramid is invertible — to reconstruct $\mathbf{x}$ from $\{\mathbf{z}^{(s)}\}_{s=1}^{S}$, we simply invert the recursive relation, letting $\mathbf{y}^{(1)} = \mathbf{z}^{(1)}$, and computing

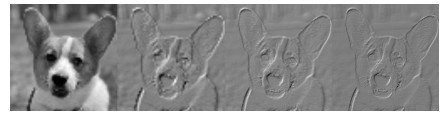

Figure 2: A Laplacian pyramid hierarchy with $S = 4$. Left to right: $z^{(1)}, \dots, z^{(4)}$.

$$\mathbf{y}^{(s+1)} = u(\mathbf{y}^{(s)}) + \mathbf{z}^{(s+1)} \tag{13}$$

for $s = 1, \dots S - 1$, where $\mathbf{x} = \mathbf{y}^{(S)}$. Finally, to see that the Laplacian pyramid is volume-preserving, we observe that it forms a tight frame (Unser et al., 2011) and therefore satisfies a generalization of Parseval's Equality, i.e., $||h(\mathbf{x})||_2^2 = ||\mathbf{x}||_2^2$. Since this holds for all $\mathbf{x}$, we can conclude that the largest and smallest singular values of $h$ are both 1.

**Wavelet Decomposition** A more efficient hierarchical encoding of $\mathbf{x}$ can be obtained by enforcing the orthogonality of $h$. While the Laplacian pyramid representation specifies a mapping into a latent space of higher dimension $\mathcal{R}^d \to \mathcal{R}^{2d}$, the wavelet transform is orthogonal (and thus $\mathcal{R}^d \to \mathcal{R}^d$). There are many types of wavelet transforms (Rioul & Vetterli, 1991; Graps, 1995), each constructed from scaled and translated versions of a respective (mother) wavelet. We focus on the Haar wavelet (Stanković & Falkowski,

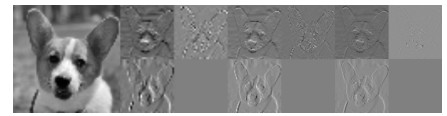

Figure 3: A wavelet hierarchy with $S = 4$. Left to right: $z^{(1)}, \dots, z^{(4)}$.

2003) as it is widely used and permits a simple implementation. In this case, the wavelet transform $\mathbf{W}$ can be further described in terms of outer products of the matrices $L = \frac{1}{\sqrt{2}}[1, 1]$ and $H = \frac{1}{\sqrt{2}}[-1, 1]$. There are four kernels total, $HH^T, HL^T, LH^T,$ and $LL^T$, each of size $2 \times 2$. Letting $*$ be the convolution operator with stride 2, the scales of the wavelet hierarchy can be written as

$$\mathbf{W}(\mathbf{y}^{(s)}) = [\underbrace{HH^T * \mathbf{y}^{(s)}}_{\mathbf{z}_{HH}^{(s)}}, \underbrace{HL^T * \mathbf{y}^{(s)}}_{\mathbf{z}_{HL}^{(s)}}, \underbrace{LH^T * \mathbf{y}^{(s)}}_{\mathbf{z}_{LH}^{(s)}}, \underbrace{LL^T * \mathbf{y}^{(s)}}_{\mathbf{z}_{LL}^{(s)}}] \quad \text{and} \quad \mathbf{y}^{(s)} = \mathbf{z}_{LL}^{(s+1)}, \tag{14}$$

for $s = 2, \dots, S$, where $\mathbf{y}^{(S)} = \mathbf{z}_{LL}^{(S+1)} := \mathbf{x}$. The $s = 2, \dots, S$th scale representations $\mathbf{z}^{(s)} = \mathbf{z}_{HH}^{(s)}, \mathbf{z}_{HL}^{(s)}, \mathbf{z}_{LH}^{(s)}$ can be seen as the high frequency subbands of $\mathbf{x}$ representing horizontal, vertical, and diagonal edge information, while $\mathbf{z}_{LL}^{(s)}$ are the low frequency residuals. Like with Laplacian pyramids, $\mathbf{z}^{(1)} := \mathbf{y}^{(1)}$ is taken to be the coarsest scale. Finally, due to its orthonormality, it is clear that this mapping satisfies the volume-preserving property of Eq. 8, and is thus a hierarchical volume-preserving map.

## 5 LIKELIHOOD TRAINING OF CASCADED DIFFUSION MODELS

In this section, we construct the probabilistic cascaded diffusion model (PCDM), which is a multi-scale hierarchical model capable of likelihood training and inference. We then investigate the theoretical properties of our model, and uncover a rigorous connection between likelihood training and score matching with respect to an optimal transport cost.

## 5.1 COUPLED DIFFUSION MODELING ON A $h$-INDUCED LATENT SPACE

Letting $h : \mathcal{X} \to \mathcal{Z}^{(1)} \times \cdots \times \mathcal{Z}^{(S)}$ be a hierarchical volume-preserving map (where $\mathbf{x} \in \mathcal{X}$ and $\mathbf{z}^{(s)} \in \mathcal{Z}^{(s)}$), we return to the cascaded diffusion model on $h(\mathbf{x}) = (\mathbf{z}^{(1)}, \mathbf{z}^{(2)}, \dots, \mathbf{z}^{(S)})$ introduced in Section 3, which is defined by way of a standard unconditional diffusion model $p_\theta(\mathbf{z}^{(1)})$ and a sequence of conditional super-resolution diffusion models $p_\theta(\mathbf{z}^{(s)}|\mathbf{z}^{(<s)})$. This produces $S$ *coupled* diffusion processes, where each scale $\mathbf{z}^{(s)}$ is governed by its own Itô SDE with dynamics that depend on the processes that precede it.

By Lemma 4.1, we see that the desired likelihood function $p_\theta(\mathbf{x})$ can be recovered by the simple relation

$$\log p_\theta(\mathbf{x}) = \log p_\theta(\mathbf{z}^{(1)}) + \sum_{s=2}^{S} \log p_\theta(\mathbf{z}^{(s)}|\mathbf{z}^{(<s)}). \tag{15}$$

Likelihood training then proceeds by lower bounding each likelihood with its corresponding loss function — $\mathcal{L}(\mathbf{z}^{(1)})$ as in Eq. 5 for the unconditional base model, and

$$\mathcal{L}(\mathbf{z}^{(s)}|\mathbf{z}^{(<s)}) = \mathbb{E}_{\mathbf{x}_{1:T}\sim q}\Bigg[ \log p_\theta(\mathbf{z}_0^{(s)}|\mathbf{z}_1^{(s)}, \mathbf{z}_0^{(<s)}) - KL(q(\mathbf{z}_T^{(s)}|\mathbf{z}_0^{(s)}, \mathbf{z}_0^{(<s)})||p_\theta(\mathbf{z}_T^{(s)}|\mathbf{z}_0^{(<s)}))$$
$$- \sum_{k=1}^{T-1} KL(q(\mathbf{z}_k^{(s)}|\mathbf{z}_{k+1}^{(s)}, \mathbf{z}_0^{(s)}, \mathbf{z}_0^{(<s)})||p_\theta(\mathbf{z}_k^{(s)}|\mathbf{z}_{k+1}^{(s)}, \mathbf{z}_0^{(<s)})) \Bigg] \tag{16}$$

for the conditional super-resolution models. Combining all terms, the full training loss of our cascaded diffusion model may be written as

$$\mathcal{C}(\mathbf{x}) = \mathcal{L}(\mathbf{z}^{(1)}) + \sum_{s=2}^{S} \mathcal{L}(\mathbf{z}^{(s)}|\mathbf{z}^{(<s)}), \tag{17}$$

over all $\mathbf{x}$ in the dataset.

## 5.2 A CONNECTION TO OPTIMAL TRANSPORT

A notable property of distances in the latent spaces induced by maps from Section 4.2 is their connection to an optimal transport cost. We begin by introducing the Wasserstein distance on the image domain.

**Definition 2** (Wasserstein $p$-Metric). *Let $\mathbf{x}, \mathbf{y} \in \mathbb{R}^{H \times W}$ be images, which we interpret as unnormalized densities on the product space of pixel indices $\{h\}_{h=1}^{H} \times \{w\}_{w=1}^{W}$. The Wasserstein $p$-metric can be written as*

$$W_p(\mathbf{x}, \mathbf{y}) = \left( \inf_{\nu} \sum_{i,j,k,\ell} \nu(i,j,k,\ell) ||\mathbf{x}_{ij} - \mathbf{y}_{k\ell}||^p \right)^{1/p}, \tag{18}$$

*where $\mathbf{x}_{ij}$ (or $\mathbf{y}_{ij}$) is the $i,j$th pixel of $\mathbf{x}$ (or $\mathbf{y}$) and $\nu(i,j,k,\ell)$ is a scalar-valued function such that $\sum_{i,j} \nu(i,j,k,\ell) = \mathbf{x}_{k\ell}$ and $\sum_{k,\ell} \nu(i,j,k,\ell) = \mathbf{x}_{ij}$.*

Our next statement builds on prior theoretical work in linear-time approximations to the Earth Mover's Distance (Shirdhonkar & Jacobs, 2008).

**Theorem 5.1** (Cascaded Diffusion Modeling and EMD Score Matching). *Let $h$ be one of the hierarchical volume-preserving maps defined in Section 4.2, and $p_\theta$ be a cascaded diffusion model on $h(\mathbf{x}) = (\mathbf{z}^{(1)}, \mathbf{z}^{(2)}, \dots, \mathbf{z}^{(S)})$. Then there exists a constant $\alpha$ depending only on the map $h$ such that*

$$\alpha\mathcal{C}(\mathbf{x}) \geq \sum_{k=1}^{T-1} w_k \mathbb{E}_{\tilde{\mathbf{x}}_k \sim q(\mathbf{x}_k|\mathbf{x}_0)} \left[ W_p(\nabla_\mathbf{x} \log q(\tilde{\mathbf{x}}_k|\mathbf{x}_0), \mathbf{s}_\theta(\tilde{\mathbf{x}}_k, t_k)) \right], \tag{19}$$

*where $w_k$ is the likelihood weighting of the variational lower bound, $q(\mathbf{x}_k|\mathbf{x}_0)$ is the marginal of the forward diffusion process, and $0 < p \leq 1$.*

| Model | CIFAR10 32x32 | ImageNet 32x32 | ImageNet 64x64 | ImageNet 128x128 |
|---|---|---|---|---|
| VAE with IAF (Kingma et al., 2016) | 3.11 | | | |
| PixelCNN (Van Den Oord et al., 2016) | 3.03 | 3.83 | 3.57 | |
| PixelCNN++ (Salimans et al., 2017) | 2.92 | | | |
| Glow (Kingma & Dhariwal, 2018) | 3.35 | 4.09 | 3.81 | |
| Image Transformer (Parmar et al., 2018) | 2.90 | 3.77 | | |
| SPN (Menick & Kalchbrenner, 2018) | | 3.52 | | |
| CR-NVAE (Sinha & Dieng, 2021) | 2.51 | | | |
| Flow++ (Ho et al., 2019a) | 3.06 | 3.86 | 3.69 | |
| Sparse Transformer (Child et al., 2019) | 2.80 | | 3.44 | |
| Sparse Transformer w/ dist. aug. (Jun et al., 2020) | 2.53 | | 3.44 | |
| Very Deep VAE (Child, 2020) | 2.87 | 3.80 | 3.52 | |
| DDPM (Ho et al., 2020) | 3.69 | | | |
| Score SDE (Song et al., 2020b) | 2.99 | | | |
| Improved DDPM (Dhariwal & Nichol, 2021) | 2.94 | | 3.54 | |
| EBM-DRL (Gao et al., 2020) | 3.18 | | | |
| Routing Transformer (Roy et al., 2021) | | | 3.43 | |
| S4 (Gu et al., 2021) | 2.85 | | | |
| Score Flow (Song et al., 2021) | 2.80 | 3.76 | | |
| LSGM (Vahdat et al., 2021) | 2.87 | | | |
| Flow Matching (Lipman et al., 2022) | 2.99 | 3.53 | 3.31 | 2.90 |
| Soft Truncation (Kim et al., 2022b) | 2.99 | 3.53 | | |
| INDM (Kim et al., 2022a) | 2.97 | | | |
| VDM (Kingma et al., 2021) | 2.49 | 3.72 | 3.40 | |
| i-DODE (Zheng et al., 2023) | 2.56 | 3.69 | | |
| *Our work* | | | | |
| LP-PCDM | 2.36 | 3.52 | 3.12 | 2.91 |
| **W-PCDM** | **2.35** | **3.32** | **2.95** | **2.64** |

Table 1: Comparison between our proposed model and other competitive models in the literature in terms of expected negative log likelihood on the test set computed as bits per dimension (BPD). Results from existing models are taken from the literature.

Theorem 5.1 reveals that optimizing Eq. 17 can also be interpreted as solving an optimal transport problem between the true and approximated scores $\nabla_{\mathbf{x}} \log q(\tilde{\mathbf{x}}_k|\mathbf{x}_0)$ and $\mathbf{s}_\theta(\tilde{\mathbf{x}}_k, t_k)$ with respect to the Wasserstein $p$-metric, for $p \in [0, 1]$. Of note is the $p = 1$ case, which is also regarded as the Earth Mover's Distance (EMD), a well-known measure in computer vision understood to correlate well with human perceptual similarity (Rubner et al., 2000). By giving up exact computation of the optimal transport cost, we are able to optimize an $\mathcal{O}(n^3 \log n)$ measure in linear time, bringing the EMD measure into the realm of feasible criteria to train diffusion models with in high dimensions.

# 6 EXPERIMENTS

We evaluate both the the Laplacian pyramid-based and wavelet-based variants of our proposed probabilistic cascading diffusion model (LP-PCDM and W-PCDM, respectively) in several settings. First, we begin on a general density estimation task on the CIFAR10 (Krizhevsky et al., 2009) and ImageNet 32, 64, and 128 (Van Den Oord et al., 2016) datasets. Then we investigate the anomaly detection performance of our models with an out-of-distribution baseline where the model is tasked to differentiate between in-distribution (CIFAR10) and out-of-distribution (SVHN (Netzer et al., 2011), uniform, and constant uniform) data. Finally, we evaluate our model on a lossless compression benchmark against several competitive baselines in neural compression literature.

## 6.1 LIKELIHOOD AND SAMPLE QUALITY

We adapt the architecture and training procedure in (Kingma et al., 2021) to our framework (Appendix B), and compare our model to existing likelihood-based generative models in terms of bits per dimension (BPD) on standard image-based density estimation datasets: CIFAR10, ImageNet 32x32,

|  | GLOW | PixelCNN++ | EBM | AR-CSM | Typ. Test | VDM | LP-PCDM | W-PCDM |
|---|---|---|---|---|---|---|---|---|
| SVHN | 0.24 | 0.32 | 0.63 | 0.68 | 0.46 | 0.53 | 0.71 | **0.74** |
| Const. Uniform | 0.0 | 0.0 | 0.30 | 0.57 | **1.0** | 0.96 | **1.0** | **1.0** |
| Uniform | **1.0** | **1.0** | **1.0** | 0.95 | **1.0** | **1.0** | **1.0** | **1.0** |
| Average | 0.41 | 0.44 | 0.64 | 0.73 | 0.81 | 0.83 | 0.90 | **0.92** |

Table 2: Out-of-distribution (OOD) detection performance of a CIFAR10 trained model on a selection of anomalous distributions in terms of AUROC. Results from existing models are taken from the literature.

and ImageNet64x64. We additionally evaluate our model on ImageNet 128x128 and compare against a higher resolution baseline (Lipman et al., 2022). Table 1 shows that we obtain state-of-the-art results across all datasets, demonstrating the advantages of density estimation with a hierarchical prior. We then evaluate the sample quality of generated images on the CIFAR10 dataset. Using the likelihood weighting scheme in (Kingma et al., 2021), we obtain an FID of 6.31 with LP-PCDM and 6.23 with W-PCDM, which improves on the FID obtained by (Kingma et al., 2021) (7.41). Switching to the architecture and weighting scheme specified in (Karras et al., 2022) optimized for sample quality, we obtain a competitive FID of 2.45 and 2.42 for LP-PCDM and W-PCDM, at the cost of increased BPD (9.55 and 10.31, respectively).

## 6.2 OUT-OF-DISTRIBUTION DETECTION

Finally, we evaluate the out-of-distribution (OOD) detection capabilities of our proposed model. In this problem, a probabilistic model is given a set of in-distribution (i.e., training) and out-of-distribution (i.e., outlier) points, and tasked with discriminating between the two distributions in an unsupervised manner. Likelihood-based models are surprisingly brittle under this benchmark (Hendrycks et al., 2018; Nalisnick et al., 2018; Choi et al., 2018; Shafaei et al., 2018), even though likelihoods $p_\theta(\mathbf{x})$ permit an intuitive interpretation for inlier/outlier classification. We use as our OOD statistic the typicality test $h_\theta(x) = |\frac{1}{M} \sum_{m=1}^{M} -\log p_\theta(\mathbf{x}_m) - \mathbb{H}(p_\theta(\mathbf{x}))|$ proposed in (Nalisnick et al., 2019), where $\log p_\theta(\mathbf{x}_m)$ is approximated by our likelihood bound $\mathcal{C}(x)$ from Eq. (17). We compute our statistic under the most difficult scenario of $M = 1$. Following (Du & Mordatch, 2019; Meng et al., 2020), we evaluate our models against the Street View House Numbers (SVHN) (Netzer et al., 2011), constant uniform, and constant distributions as OOD distributions. Table 2 shows that we obtain significant gains in OOD performance. More experimental details are in Appendix B.1.

## 6.3 LOSSLESS PROGRESSIVE CODING

As discussed in (Ho et al., 2020; Kingma et al., 2021; Hoogeboom et al., 2021), a probabilistic diffusion model can also be seen as a latent variable model in a neural network-based lossless compression scheme. We implement a neural compressor using our proposed model as the latent variable model in the Bits-Back ANS algorithm (Townsend et al., 2019), and compare against several prominent models in the field, demonstrating significant reductions in compression size on the CIFAR10 dataset. We

| Model | Compression BPD |
|---|---|
| FLIF (Sneyers & Wuille, 2016) | 4.14 |
| IDF (Hoogeboom et al., 2019) | 3.26 |
| LBB (Ho et al., 2019b) | 3.12 |
| VDM (Kingma et al., 2021) | 2.72 |
| ARDM (Hoogeboom et al., 2021) | 2.71 |
| LP-PCDM | 2.40 |
| **W-PCDM** | **2.37** |

Table 3: Lossless compression performance on CIFAR10 in terms of bits per dimension (BPD). Results from existing models are taken from the literature.

report performance in terms of average bits per dimension of the codelength over the entire dataset in Table 3. One downside of a bits back based implementation is that there is a significant overhead of bits required to initialize the algorithm, and encoding and decoding must be performed in a fixed sequence decided at encoding-time, meaning that single image compression is expensive. This limitation can be removed by implementing a direct compressor (Hoogeboom et al., 2021), which can also be applied to our model. We leave this avenue of research for further work.

## 7 CONCLUSION AND FUTURE DIRECTIONS

We demonstrated the existence of a class of transformations we call hierarchical volume-preserving maps that sidesteps a common difficulty in multi-scale likelihoood modeling, the intractability of the likelihood function $p_\theta(\mathbf{x})$. Hierarchical volume-preserving maps are homeomorphisms to which the likelihood function are invariant, allowing the likelihood function $p_\theta(\mathbf{x})$ to be directly computed as a joint likelihood on the scales. We then presented two particular instances of these transformations that warrant attention due to their simplicity and performance on empirical benchmarks. Finally, we demonstrate a connection between our training and optimal transport. An open question is whether score matching under an EMD norm enjoys the same statistical guarantees as the standard score matching, e.g., consistency, efficiency, and asymptotic normality (Hyvärinen, 2006; Song et al., 2020a). We hope that this work paves the way for future inroads between hierarchical and likelihood-based modeling, and that our theoretical framework opens the door for the further diversity and improvements in the design of diffusion models.

## 8 ACKNOWLEDGEMENTS

YK acknowledges support for the research of this work from NIH [R01GM131642, UM1PA051410, U54AG076043, U54AG079759, P50CA121974, U01DA053628 and R33DA047037]. RB's research at the Weizmann Institute was supported partially by the Israel Science Foundation, grant No. 1639/19, by the Israeli Council for Higher Education (CHE) via the Weizmann Data Science Research Center, by the MBZUAI-WIS Joint Program for Artificial Intelligence Research, and by research grants from the Estates of Tully and Michele Plesser and the Anita James Rosen and Harry Schutzman Foundations.

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

## A  PROOFS

**Lemma 4.1** (Probabilistic Invariance of Hierarchical Volume-preserving Maps). *Let $h$ be a hierarchical volume-preserving map such that $h(\mathbf{x}) = (\mathbf{z}^{(1)}, \mathbf{z}^{(2)}, \ldots, \mathbf{z}^{(S)})$, and $p_\theta$ be a likelihood function on $\mathbf{z}^{(1)}, \mathbf{z}^{(2)}, \ldots, \mathbf{z}^{(S)}$. Then the likelihood function with respect to the original data $p_\theta(\mathbf{x})$ can be recovered by the simple relation*

$$\log p_\theta(\mathbf{x}) = \log p_\theta[h(\mathbf{x})]. \tag{9}$$

*Proof.* We begin with the generalized change-of-variables formula on Riemannian manifolds (Ben-Israel, 1999), which relates the density function on $\mathbf{x}$ with another on $\mathbf{y} = f(\mathbf{x})$ under the (potentially manifold-valued) transformation $f$ via the relation

$$p(\mathbf{x}) = p(\mathbf{y}) \sqrt{\det\left(\left[\frac{\partial}{\partial \mathbf{x}} f(\mathbf{x})\right]^T \left[\frac{\partial}{\partial \mathbf{x}} f(\mathbf{x})\right]\right)}. \tag{20}$$

Considering now the likelihood function $p_\theta$, letting $f = h$, and taking logs of both sides, we see that

$$\log p_\theta(\mathbf{x}) = \log p_\theta(\mathbf{z}^{(1)}, \mathbf{z}^{(2)}, \ldots, \mathbf{z}^{(S)}) + \log \sqrt{\det\left(\left[\frac{\partial}{\partial x} h(\mathbf{x})\right]^T \left[\frac{\partial}{\partial x} h(\mathbf{x})\right]\right)}$$

$$= \log p_\theta(\mathbf{z}^{(1)}, \mathbf{z}^{(2)}, \ldots, \mathbf{z}^{(S)}) + \log |1|$$

$$= \log p_\theta(h(\mathbf{x})),$$

where the second equality follows due to the fact that $h$ is volume-preserving. We have thus shown that the likelihood function $p_\theta(\mathbf{x})$ coincides with the joint likelihood $p_\theta(\mathbf{z}^{(1)}, \mathbf{z}^{(2)}, \ldots, \mathbf{z}^{(S)})$. This concludes the proof. $\square$

**Theorem 5.1** (Cascaded Diffusion Modeling and EMD Score Matching). *Let $h$ be one of the hierarchical volume-preserving maps defined in Section 4.2, and $p_\theta$ be a cascaded diffusion model on $h(\mathbf{x}) = (\mathbf{z}^{(1)}, \mathbf{z}^{(2)}, \ldots, \mathbf{z}^{(S)})$. Then there exists a constant $\alpha$ depending only on the map $h$ such that*

$$\alpha\mathcal{C}(\mathbf{x}) \geq \sum_{k=1}^{T-1} w_k \mathbb{E}_{\tilde{\mathbf{x}}_k \sim q(\mathbf{x}_k|\mathbf{x}_0)}\left[W_p(\nabla_{\mathbf{x}} \log q(\tilde{\mathbf{x}}_k|\mathbf{x}_0), \mathbf{s}_\theta(\tilde{\mathbf{x}}_k, t_k))\right], \tag{19}$$

*where $w_k$ is the likelihood weighting of the variational lower bound, $q(\mathbf{x}_k|\mathbf{x}_0)$ is the marginal of the forward diffusion process, and $0 < p \leq 1$.*

*Proof.* According to (Ho et al., 2020), the likelihood bound Eq. 5

$$\mathcal{L}(\mathbf{z}^{(1)}) \geq \mathbb{E}_{\mathbf{x}_{1:T} \sim q}\bigg[\underbrace{\log p_\theta(\mathbf{z}_0^{(1)}|\mathbf{z}_1^{(1)})}_{\mathcal{L}_0(\mathbf{z}^{(1)})} - \underbrace{KL(q(\mathbf{z}_T^{(1)}|\mathbf{z}_0^{(1)})||p_\theta(\mathbf{z}_T^{(1)}))}_{\mathcal{L}_T(\mathbf{z}^{(1)})}$$

$$- \sum_{k=1}^{T-1} \underbrace{\log KL(q(\mathbf{z}_k^{(1)}|\mathbf{z}_{k+1}^{(1)}, \mathbf{z}_0^{(1)})||p_\theta(\mathbf{z}_k^{(1)}|\mathbf{z}_{k+1}^{(1)}))}_{\mathcal{L}_k(\mathbf{z}^{(1)})}\bigg], \tag{21}$$

can be decomposed into the three elements: a decoder reconstruction term $\mathcal{L}_0(\mathbf{x})$, a prior loss term $\mathcal{L}_T(\mathbf{x})$, and a weighted sum over the diffusion score matching terms $\sum_{k=1}^{T-1} \mathcal{L}_k(\mathbf{x})$. The conditional likelihood bound Eq. 16 can be decomposed in a similar fashion:

$$\mathcal{L}(\mathbf{z}^{(s)}|\mathbf{z}^{(<s)}) \geq \mathbb{E}_{\mathbf{x}_{1:T} \sim q}\bigg[\underbrace{\log p_\theta(\mathbf{z}_0^{(s)}|\mathbf{z}_1^{(s)}, \mathbf{z}_0^{(<s)})}_{\mathcal{L}_0(\mathbf{z}^{(s)}|\mathbf{z}^{(<s)})} - \underbrace{KL(q(\mathbf{z}_T^{(s)}|\mathbf{z}_0^{(s)})||p_\theta(\mathbf{z}_T^{(s)}|\mathbf{z}_0^{(<s)}))}_{\mathcal{L}_T(\mathbf{z}^{(s)}|\mathbf{z}^{(<s)})}$$

$$- \sum_{k=1}^{T-1} \underbrace{\log KL(q(\mathbf{z}_k^{(s)}|\mathbf{z}_{k+1}^{(s)}, \mathbf{z}_0^{(s)}, \mathbf{z}_0^{(<s)})||p_\theta(\mathbf{z}_k^{(s)}|\mathbf{z}_{k+1}^{(s)}, \mathbf{z}_0^{(<s)}))}_{\mathcal{L}_k(\mathbf{z}^{(s)})|\mathbf{z}^{(<s)}}\bigg], \tag{22}$$

for $s = 2, \ldots, S$.

Under the standard Gaussian assumption on the marginals of the diffusion process (i.e., diffusion process is a Gaussian process), the unconditional diffusion score matching terms can also be written as

$$\mathcal{L}_k(\mathbf{z}^{(1)}) = w_k \mathbb{E}_{\boldsymbol{\epsilon} \sim \mathcal{N}(0,\mathbf{I})}[||\boldsymbol{\epsilon} - \boldsymbol{\epsilon}_\theta(\tilde{\mathbf{z}}_k^{(1)}, t_k)||^2], \quad \text{where} \quad \tilde{\mathbf{z}}_k^{(1)} = \sqrt{1-\beta}\mathbf{z}^{(1)} + \sqrt{\beta}\boldsymbol{\epsilon}, \quad (23)$$

$\beta$ is the standard deviation of the marginal of the diffusion process at time $t$, and $w_k$ is the likelihood weighting of the score matching loss. Since the epsilon network can be related to the score network by the relation $\boldsymbol{\epsilon}_\theta(\mathbf{x}, t) = \sqrt{\beta}\mathbf{s}_\theta(\mathbf{x}, t)$, we can rewrite the above equation as

$$\mathcal{L}_k(\mathbf{z}^{(1)}) = w_k \mathbb{E}_{\boldsymbol{\epsilon} \sim \mathcal{N}(0,\mathbf{I})}[||\frac{\boldsymbol{\epsilon}}{\sqrt{\beta}} - \mathbf{s}_\theta(\tilde{\mathbf{z}}_k^{(1)}, t_k)||^2]$$

$$= w_k \mathbb{E}_{\boldsymbol{\epsilon} \sim \mathcal{N}(0,\mathbf{I})}[||\nabla_{\mathbf{z}^{(1)}} \log q(\tilde{\mathbf{z}}_k^{(1)}|\mathbf{z}_0^{(1)}) - \mathbf{s}_\theta(\tilde{\mathbf{z}}_k^{(1)}, t_k)||^2],$$

where the second equality is due to the connection between denoising and score matching discussed in (Vincent, 2011; Ho et al., 2020). Again, the conditional diffusion score matching loss can be written similarly:

$$\mathcal{L}_k(\mathbf{z}^{(s)}|\mathbf{z}^{(<s)}) = w_k \mathbb{E}_{\boldsymbol{\epsilon} \sim \mathcal{N}(0,\mathbf{I})}[||\nabla_{\mathbf{z}^{(s)}} \log q(\tilde{\mathbf{z}}_k^{(s)}|\mathbf{z}_0^{(s)}) - \mathbf{s}_\theta(\tilde{\mathbf{z}}_k^{(1)}, \mathbf{z}^{(<s)}, t_k)||^2], \quad (24)$$

for $s = 2, \ldots, S$.

Simplifying notation, we denote the score networks as $\mathbf{s}_\theta^{(1)} := \mathbf{s}_\theta(\tilde{\mathbf{z}}_k^{(1)}, t_k)$ and $\mathbf{s}_\theta^{(s)} := \mathbf{s}_\theta(\tilde{\mathbf{z}}_k^{(s)}, \mathbf{z}^{(<s)}, t_k)$. Since the same noise schedule is used across the scales of the cascading model, we may combine all diffusion score matching losses to obtain

$$\mathcal{C}_k := \mathcal{L}_k(\mathbf{z}^{(1)}) + \sum_{s=2}^{S} \mathcal{L}_k(\mathbf{z}^{(s)}|\mathbf{z}^{(<s)}) \quad (25)$$

$$= w_k \sum_{s=1}^{S} \mathbb{E}_{\boldsymbol{\epsilon} \sim \mathcal{N}(0,\mathbf{I})}[||\nabla_{\mathbf{z}^{(s)}} \log q(\tilde{\mathbf{z}}_k^{(s)}|\mathbf{z}_0^{(s)}) - \mathbf{s}_\theta^{(s)}||^2]. \quad (26)$$

Finally, when $h$ is the wavelet transform, Lemma A.2 provides the existence of a constant $\alpha > 0$ such that

$$\alpha w_k \sum_{s=1}^{S} \mathbb{E}_{\boldsymbol{\epsilon} \sim \mathcal{N}(0,\mathbf{I})}[||\nabla_{\mathbf{z}^{(s)}} \log q(\tilde{\mathbf{z}}_k^{(s)}|\mathbf{z}_0^{(s)}) - \mathbf{s}_\theta^{(s)}||^2]$$

$$\geq w_k \mathbb{E}_{\tilde{\mathbf{x}}_k \sim q(\mathbf{x}_k|\mathbf{x}_0)}[W_p(\nabla_{\mathbf{x}} \log q(\tilde{\mathbf{x}}_k|\mathbf{x}_0), \mathbf{s}_\theta(\tilde{\mathbf{x}}_k, t_k))]. \quad (27)$$

And since $\mathcal{L}_0$ and $\mathcal{L}_T$ are both nonnegative, we have our desired result

$$\alpha \mathcal{C}(\mathbf{x}) \geq w_k \mathbb{E}_{\tilde{\mathbf{x}}_k \sim q(\mathbf{x}_k|\mathbf{x}_0)}[W_p(\nabla_{\mathbf{x}} \log q(\tilde{\mathbf{x}}_k|\mathbf{x}_0), \mathbf{s}_\theta(\tilde{\mathbf{x}}_k, t_k))]. \quad (28)$$

To obtain this result when $h$ is the Laplacian pyramid mapping, we note that under our choice of downscaling and upscaling functions $d(\cdot)$ and $u(\cdot)$ (the norm preserving bilinear resampling method), the auxiliary variables $\mathbf{y}$ (Eq. 12) at each scale $s$ are also those of the wavelet hierarchy (Eq. 14). This is because the $LL^T$ convolution kernel with stride 2 is the bilinear downsampling operator. Thus $d(\cdot) = LL^T * (\cdot)$. Now, since the wavelet operator $\mathbf{W}$, $d(\cdot)$ and $u(\cdot)$ are norm preserving,

$$||\mathbf{z}_{\text{lp}}^{(s)}||_2 = ||\mathbf{y}_{\text{lp}}^{(s)} - u(d(\mathbf{y}^{(s)}))||_2$$

$$= ||\mathbf{y}_{\text{w}}^{(s)} - u(\mathbf{y}_{\text{w}}^{(s-1)})||_2$$

$$= ||\mathbf{W}[\mathbf{y}_{\text{w}}^{(s)} - u(\mathbf{y}_{\text{w}}^{(s-1)})]||_2$$

$$= ||[\mathbf{z}_{HH}^{(s)}, \mathbf{z}_{HL}^{(s)}, \mathbf{z}_{LH}^{(s)}, \mathbf{z}_{LL}^{(s)}] - [0, 0, 0, \mathbf{z}_{LL}^{(s)}]||_2$$

$$= ||\mathbf{z}_{\text{w}}^{(s)}||_2.$$

Therefore, the norms of the scales coincide at each level for all $\mathbf{x}$, and we can conclude that the Laplacian pyramid diffusion score matching loss

$$\mathcal{C}_k = w_k \sum_{s=1}^{S} \mathbb{E}_{\boldsymbol{\epsilon} \sim \mathcal{N}(0,\mathbf{I})}[||\nabla_{\mathbf{z}^{(s)}} \log q(\tilde{\mathbf{z}}_k^{(s)}|\mathbf{z}_0^{(s)}) - \mathbf{s}_\theta^{(s)}||^2]$$

coincides with that of the wavelet loss [2]. Finally, we invoke Lemma A.2 to obtain the desired result.

$\square$

To show Lemma A.2, we first restate a key result from (Shirdhonkar & Jacobs, 2008) using the notation in this text.

**Lemma A.1** (From Theorem 2 in (Shirdhonkar & Jacobs, 2008)). *Consider the optimal transport cost $W_p(\mathbf{x}, \mathbf{y})$ (Eq. 2) for $p \in [0, 1]$ between two unnormalized histograms $\mathbf{x}$ and $\mathbf{y}$. Let $h(\mathbf{x} - \mathbf{y}) = (\mathbf{z}^{(1)}, \mathbf{z}^{(2)}, \dots, \mathbf{z}^{(S)})$, and*

$$\hat{\mu} = C_0 ||\mathbf{z}^{(1)}||_1 + C_1 \sum_{s=2}^{S} 2^{-s(p+n/2)} ||\mathbf{z}^{(s)}||_1, \tag{29}$$

*where $n$ is maximum height / width of $\mathbf{x}$ and $\mathbf{y}$. $\mathbf{z}^{(1)}$ and $\{\mathbf{z}^{(s)}\}_{s=2}^{S}$ are the detail and approximation coefficients of the wavelet transform respectively. Then there exist positive constants $C_L, C_U$ such that*

$$C_L \hat{\mu} \leq W_p(\mathbf{x}, \mathbf{y}) \leq C_U \hat{\mu}. \tag{30}$$

Now we are able to make the following statement.

**Lemma A.2.** *Consider the optimal transport cost $W_p(\cdot, \cdot)$ (Eq. 2) for $p \in [0, 1]$ between the true and modeled score functions $\nabla_{\mathbf{x}} \log q(\tilde{\mathbf{x}}_k | \mathbf{x}_0)$ and $\mathbf{s}_\theta(\tilde{\mathbf{x}}_k, k)$. Let $h$ be a hierarchical volume-preserving map,*

$$h(\nabla_{\mathbf{x}} \log q(\tilde{\mathbf{x}}_k | \mathbf{x}_0)) = (\nabla_{\mathbf{z}^{(1)}} \log q(\tilde{\mathbf{z}}_k^{(1)} | \mathbf{z}_0^{(1)}), \nabla_{\mathbf{z}^{(2)}} \log q(\tilde{\mathbf{z}}_k^{(2)} | \mathbf{z}_0^{(2)}), \dots, \nabla_{\mathbf{z}^{(S)}} \log q(\tilde{\mathbf{z}}_k^{(S)} | \mathbf{z}_0^{(S)})), \tag{31}$$

*and*

$$h(\mathbf{s}_\theta(\tilde{\mathbf{x}}_k, k)) = (\mathbf{s}_\theta(\tilde{\mathbf{z}}_k^{(1)} | t_k), \mathbf{s}_\theta(\tilde{\mathbf{z}}_k^{(2)} | \mathbf{z}^{(1)}, t_k), \dots, \mathbf{s}_\theta(\tilde{\mathbf{z}}_k^{(S)} | \mathbf{z}^{(<S)}, t_k)). \tag{32}$$

*Then there exists a constant $\beta > 0$ such that*

$$W_p(\nabla_{\mathbf{x}} \log q(\tilde{\mathbf{x}}_k | \mathbf{x}_0), \mathbf{s}_\theta(\tilde{\mathbf{x}}_k)) \leq \beta \sum_{s=1}^{S} ||\nabla_{\mathbf{z}^{(s)}} \log q(\tilde{\mathbf{z}}_k^{(s)} | \mathbf{z}_0^{(s)}) - \mathbf{s}_\theta(\mathbf{x})||_2. \tag{33}$$

*Proof.* Note that for any scale level $1 \leq s \leq S$, image size $n \geq 1$, and $p \in [0, 1]$, we have that $s(p + n/2) \geq 0$ and thus $2^{-s(p+n/2)} \leq 1$. This allows us to upper bound the right side inequality in

---

[2]While the losses are the same for both models for each $\mathbf{x}$, the gradients and therefore training trajectories are very different due to the different parameterizations of the representation space. This is why we get differing behaviors between the two hierarchical maps.

Eq. 30 as

$$W_p(\nabla_{\mathbf{x}} \log q(\tilde{\mathbf{x}}_k|\mathbf{x}_0), \mathbf{s}_\theta(\tilde{\mathbf{x}}_k, t_k)) \le C_U \Bigg( C_0 ||\nabla_{\mathbf{z}^{(1)}} \log q(\tilde{\mathbf{z}}_k^{(1)}|\mathbf{z}_0^{(1)}) - \mathbf{s}_\theta(\mathbf{z}^{(1)}|t_k)||_1 \tag{34}$$

$$+ C_1 \sum_{s=2}^{S} 2^{-s(p+n/2)} ||\nabla_{\mathbf{z}^{(1)}} \log q(\tilde{\mathbf{z}}_k^{(s)}|\mathbf{z}_0^{(s)}) \tag{35}$$

$$- \mathbf{s}_\theta(\mathbf{z}^{(s)}|\mathbf{z}^{(<s)}, t_k)||_1 \Bigg) \tag{36}$$

$$\le C_U \Bigg( C_0 ||\nabla_{\mathbf{z}^{(1)}} \log q(\tilde{\mathbf{z}}_k^{(1)}|\mathbf{z}_0^{(1)}) - \mathbf{s}_\theta(\mathbf{z}^{(1)}|t_k)||_1 \tag{37}$$

$$+ C_1 \sum_{s=2}^{S} ||\nabla_{\mathbf{z}^{(1)}} \log q(\tilde{\mathbf{z}}_k^{(s)}|\mathbf{z}_0^{(s)}) \tag{38}$$

$$- \mathbf{s}_\theta(\mathbf{z}^{(s)}|\mathbf{z}^{(<s)}, t_k)||_1 \Bigg) \tag{39}$$

$$\le C_U \Bigg( \max(C_0, C_1) \sum_{s=1}^{S} ||\nabla_{\mathbf{z}^{(s)}} \log q(\tilde{\mathbf{z}}_k^{(s)}|\mathbf{z}_0^{(s)}) \tag{40}$$

$$- \mathbf{s}_\theta(\mathbf{z}^{(s)}|\mathbf{z}^{(<s)}, t_k)||_1 \Bigg). \tag{41}$$

Applying the Cauchy-Schwarz bound to the norms and collecting all constants into $\beta$, we obtain the desired inequality. □

## B MODEL AND IMPLEMENTATION

All training is performed on 8x NVIDIA RTX A6000 GPUs. We construct our cascaded diffusion models with antithetic time sampling and a learnable noise schedule as in (Kingma et al., 2021). Our noise prediction networks $\epsilon_\theta$ also follow closely to the VDM U-Net implementation in (Kingma et al., 2021), differing only in the need to deal with the representations induced by the hierarchical volume-preserving map $h$. A hierarchical model with $S$ scales is composed of $S$ separate noise prediction networks. For each input $\mathbf{x}$, our model decomposes $\mathbf{x}$ into its latent scales $\{\mathbf{z}^{(s)}\}_{s=1}^{S}$ via $h$ and directs each $\mathbf{z}^{(s)}$ to its corresponding noise prediction network $\epsilon_\theta(\mathbf{z}^{(s)}|\mathbf{z}^{(<s)}, t)$. The noise prediction network can be related to the approximated score function via

$$\epsilon_\theta(\mathbf{z}^{(s)}|\mathbf{z}^{(<s)}, t) = \sigma(t)s_\theta(\mathbf{z}^{(s)}|\mathbf{z}^{(<s)}, t), \tag{42}$$

where $\sigma(t)$ is the variance of the diffusion process at time $t$. The base noise prediction network is constructed entirely identically to the VDM architecture, as it does not rely on any additional input. The conditional noise prediction networks at each subsequent scale additionally incorporate the conditioning information from previous scales (i.e., $\mathbf{z}^{(<s)}$) via a channel-wise concatenation of the conditional information to the input, resulting in the full input vector

$$\mathbf{z}_{input}^{(s)} = [\mathbf{z}^{(s)}, \mathbf{z}_{cond}^{(<s)}] \tag{43}$$

$$\mathbf{z}_{cond}^{(<s)} = \underbrace{(d \circ d \circ \cdots \circ d)}_{(S-s+1) \text{ times}}(h^{-1}(\mathbf{z}^{(<s)}, \underbrace{\mathbf{0}}_{\in \mathcal{Z}^{(s)}}, \underbrace{\mathbf{0}}_{\in \mathcal{Z}^{(s+1)}}, \ldots, \underbrace{\mathbf{0}}_{\in \mathcal{Z}^{(S)}})) \tag{44}$$

where $h^{-1}$ is the inverse of the hierarchical volume-preserving map, $d()$ is a downsampling operator (Section 4.2), and $\mathbf{z}_{cond}^{(<s)}$ can be seen as the reconstructed image containing *only* the conditioning information available at scale $s$. Thus the noise prediction model at each scale $\epsilon_\theta(\mathbf{z}^{(s)}|\mathbf{z}^{(<s)}, t)$ takes as input $\mathbf{z}_{input}^{(s)}$ and outputs $\mathbf{z}^{(s)}$.

In the case where $h$ is the wavelet transform, the scales $\mathbf{z}^{(<s)}$ can directly be used to construct an image with the same extent as $\mathbf{z}^{(s)}$, resembling a downsampled version of the original input $\mathbf{x}$

comprising of all frequency information up to, but not including, the present scale. In this case, the reconstructed image may be directly concatenated channel-wise to the $s$-level wavelet coefficients, producing an image with 12 channels.

When $h$ is the Laplacian pyramid map, the scales $\mathbf{z}^{(<s)}$ instead form an image exactly half the size of the current representation $\mathbf{z}^{(s)}$, and therefore must be upscaled. Since the representation at each scale always has 3 dimensions (as oppposed to 9 with wavelet coefficients), the final image has 6 channels.

For CIFAR10, we use two scales for both LP-PCDM and W-PCDM, resulting in representations at each scale of
$$[16 \times 16 \times 3, 32 \times 32 \times 3]$$
for Laplacian pyramids and
$$[16 \times 16 \times 3, 16 \times 16 \times 9]$$
for the wavelet transform. We use a U-Net of depth 32, consisting of 32 residual blocks in the forward and reverse directions, respectively. We additionally incorporate a single attention layer and two additional residual blocks in the middle. We use a convolution dimension 128 in the base model and 64 in the conditional super-resolution model, resulting in 35M and 42M parameter models, respectively. We train with AdamW for 2 million updates.

For ImageNet32, we again use two scales for both LP-PCDM and W-PCDM, resulting in representations at each scale of
$$[16 \times 16 \times 3, 32 \times 32 \times 3]$$
for Laplacian pyramids and
$$[16 \times 16 \times 3, 16 \times 16 \times 9]$$
for the wavelet transform. We again use a U-Net of depth 32, consisting of 32 residual blocks in the forward and reverse directions, and incorporate a single attention layer with two additional residual blocks in the middle. We use 256 channels for the base model and 128 channels in the conditional super-resolution, resulting in 65M and 70M parameter models. We train with AdamW for 10 million updates.

For ImageNet64, we use three scales for both LP-PCDM and W-PCDM, resulting in representations at each scale of
$$[16 \times 16 \times 3, 32 \times 32 \times 3, 64 \times 64 \times 3]$$
for Laplacian pyramids and
$$[16 \times 16 \times 3, 16 \times 16 \times 9, 32 \times 32 \times 9]$$
for the wavelet transform. We now use a U-Net of depth 64, and incorporate a single attention layer with two additional residual blocks in the middle. We use 256 channels for the base model and 128 channels in the conditional super-resolution models, resulting in models with 299M and 329M parameters, respectively. We train with AdamW for 10 million updates.

Finally, for ImageNet128, we use four scales for both LP-PCDM and W-PCDM, resulting in representations at each scale of
$$[16 \times 16 \times 3, 32 \times 32 \times 3, 64 \times 64 \times 3, 128 \times 128 \times 3]$$
for Laplacian pyramids and
$$[16 \times 16 \times 3, 16 \times 16 \times 9, 32 \times 32 \times 9, 64 \times 64 \times 9]$$
for the wavelet transform. We again use a U-Net of depth 64, and incorporate a single attention layer with two additional residual blocks in the middle. We use 256 channels for the base model and 128 channels in the conditional super-resolution models, resulting in models with 400M and 440M parameters, respectively. We train with AdamW for 10 million updates.

For out-of-Distribution (OOD) detection in Section 6.2, we approximate the summation $\sum_k \mathcal{L}_k$ in the likelihood bound with an $N = 20$ sample Monte Carlo estimate, rather computing the full sum. We find empirically that this is sufficient for OOD detection. Moreover, OOD performance only improves with larger $N$. Therefore $N = 20$ was chosen as a suitable trade-off between performance and runtime complexity.

### B.1 OUT-OF-DISTRIBUTION DETECTION EXPERIMENTAL DETAILS

As proposed in (Hendrycks et al., 2018; Nalisnick et al., 2018; Choi et al., 2018; Shafaei et al., 2018; Du & Mordatch, 2019; Meng et al., 2020), a benchmark for evaluating the anomaly detection capabilities of a likelihood model $p_\theta(x)$ trained on true data $x \sim p(x)$ can be framed as a binary classification task between in-distribution $x_{in} \sim p(x)$ and out-of-distribution $x_{out} \sim \nu(x)$ data. Namely, we define data-label pairs $\{x_{in,i}, 0\}_{i=1}^{N_{in}} \cup \{x_{out,j}, 1\}_{j=1}^{N_{out}}$ where $N_in = N_out$. (Du & Mordatch, 2019; Meng et al., 2020) consider three different "out-of-distribution" densities, the $x_{out}$ drawn from the Street View House Numbers (SVHN) dataset, the distribution of constant zeros (Const. Uniform), i.e., $\nu = \delta_0$ is the Dirac delta at the all-zeros vector, and the uniform distribution, i.e., $\nu = \text{Unif}([-1,1])$. The rows of Table 2 show classification performance under the AUROC operator.

## C  A VARIATIONAL LOWER BOUND WITHOUT THE VOLUME-PRESERVING PROPERTY

Our proposed hierarchical volume-preserving maps retain **exact** likelihood evaluation capabilities in all cascaded diffusion models with latent scales satisfying $h(\mathbf{x}) = (\mathbf{z}^{(1)}, \mathbf{z}^{(2)}, , \ldots, \mathbf{z}^{(S)})$, where $h$ is a hierarchical volume-preserving map satisfying Eq. 8. On the other hand, when one does not need exact likelihoods — e.g., simply a lower bound on the likelihood is sufficient — an alternative method for obtaining a tractable loss can be obtained by folding the latent variables into the variational bound:

$$\log p_\theta(\mathbf{x}) \geq \mathbb{E}_{\mathbf{x}_{1:T} \sim q}[\log p_\theta(\mathbf{x}, \mathbf{z}) - \log q(\mathbf{z}|\mathbf{x})]$$

$$= \mathbb{E}_{\mathbf{x}_{1:T} \sim q}[\log p_\theta(\mathbf{x}, \mathbf{z}_{0:T}^{(1)}, \ldots, \mathbf{z}_{0:T}^{(S)}) - \log q(\mathbf{z}_{1:T}^{(1)}, \ldots, \mathbf{z}_{1:T}^{(S)}|\mathbf{z}_0^{(1)}, \ldots, \mathbf{z}_0^{(S)}, \mathbf{x})]$$

$$= \mathbb{E}_{\mathbf{x}_{1:T} \sim q}\left[ \underbrace{\log p_\theta(\mathbf{x}|\mathbf{z}_0^{(1)}, \ldots, \mathbf{z}_0^{(S)})}_{=0} + \sum_{s=1}^{S} \underbrace{\log q(\mathbf{z}_0^{(1)}, \ldots, \mathbf{z}_0^{(S)})|\mathbf{x})}_{=0} \right.$$
$$\left. + \log p_\theta\left(\mathbf{z}_{0:T}^{(s)}|\mathbf{z}_0^{(<s)}\right) - \log q\left(\mathbf{z}_{1:T}^{(s)}|\mathbf{z}_0^{(\leq s)}, \mathbf{x}\right) \right]$$

$$= \sum_{s=1}^{S} \mathbb{E}_{\mathbf{x}_{1:T} \sim q}\left[ \log p_\theta\left(\mathbf{z}_0^{(s)}|\mathbf{z}_1^{(s)}, \mathbf{z}_0^{(<s)}\right) + \log p_\theta\left(\mathbf{z}_{1:T}^{(s)}|\mathbf{z}_0^{(<s)}\right) - \log q\left(\mathbf{z}_{1:T}^{(s)}|\mathbf{z}_0^{(\leq s)}, \mathbf{x}\right) \right]$$

$$= \sum_{s=1}^{S} \mathbb{E}_{\mathbf{x}_{1:T} \sim q}\left[ \log p_\theta\left(\mathbf{z}_0^{(s)}|\mathbf{z}_1^{(s)}, \mathbf{z}_0^{(<s)}\right) + KL\left(p_\theta\left(\mathbf{z}_T^{(s)}|\mathbf{z}_0^{(<s)}\right) ||q\left(\mathbf{z}_{1:T-1}^{(s)}|\mathbf{z}_0^{(\leq s)}, \mathbf{x}\right)\right) \right.$$
$$\left. + KL\left(p_\theta\left(\mathbf{z}_{1:T-1}^{(s)}|\mathbf{z}_0^{(<s)}\right) ||q\left(\mathbf{z}_{1:T-1}^{(s)}|\mathbf{z}_0^{(\leq s)}, \mathbf{x}\right)\right) \right]$$

$$= \sum_{s=1}^{S} \mathbb{E}_{\mathbf{x}_{1:T} \sim q}\left[ \log p_\theta(\mathbf{z}_0^{(s)}|\mathbf{z}_1^{(s)}, \mathbf{z}_0^{(<s)}) - KL(q(\mathbf{z}_T^{(s)}|\mathbf{z}_0^{(s)}, \mathbf{z}_0^{(<s)})||p_\theta(\mathbf{z}_T^{(s)}|\mathbf{z}_0^{(<s)})) \right.$$
$$\left. - \sum_{k=1}^{T-1} KL(q(\mathbf{z}_k^{(s)}|\mathbf{z}_{k+1}^{(s)}, \mathbf{z}_0^{(s)}, \mathbf{z}_0^{(<s)})||p_\theta(\mathbf{z}_k^{(s)}|\mathbf{z}_{k+1}^{(s)}, \mathbf{z}_0^{(<s)})) \right]$$

This produces a similar form to our loss function, with the exception being that $h$ is not volume-preserving. However, this bound is much looser than our proposed bound. We denote as C-VDM a cascading diffusion model (Kingma et al., 2021) designed with this standard non-volume-preserving hierarchical representation (Ho et al., 2022) (i.e., Section 4.1). All model architecture and hyper-parameters are retained from W-PCDM and LP-PCDM, except for the choice of $\mathbf{z}^{(1)}, \ldots, \mathbf{z}^{(S)}$. In Tables 4, 5, and 6, we see that the removing volume-preservation property of $h$ significantly degrades likelihood modeling performance. We theorize that this is due to the looser bound obtained by the variational bound derivation, versus the exact equality afforded by Lemma 4.1.

| Model | CIFAR10 32x32 | ImageNet 32x32 | ImageNet 64x64 | ImageNet 128x128 |
|---|---|---|---|---|
| C-VDM | 3.20 | 4.21 | 3.87 | 3.75 |
| *Our work* | | | | |
| LP-PCDM | 2.36 | 3.52 | 3.12 | 2.91 |
| **W-PCDM** | **2.35** | **3.32** | **2.95** | **2.64** |

Table 4: Ablative study illustrating the importance of the volume-preserving property on a density estimation benchmark. Results are reported in terms of expected negative log likelihood on the test set computed as bits per dimension (BPD). Details in Appendix C.

| VDM | C-VDM | LP-PCDM | W-PCDM |
|---|---|---|---|
| SVHN | 0.43 | 0.71 | **0.74** |
| Const. Uniform | 0.55 | **1.0** | **1.0** |
| Uniform | **1.0** | **1.0** | **1.0** |
| Average | 0.66 | 0.90 | **0.92** |

Table 5: Ablative study illustrating the importance of the volume-preserving property on performance in an out-of-distribution (OOD) benchmark of a CIFAR10 trained model on a selection of anomalous distributions in terms of AUROC. Results from existing models are taken from the literature. Details in Appendix C.

| Model | Compression BPD |
|---|---|
| C-VDM | 3.22 |
| LP-PCDM | 2.40 |
| **W-PCDM** | **2.37** |

Table 6: Ablative study illustrating the importance of the volume-preserving property on lossless compression performance on CIFAR10. Results reported in terms of bits per dimension (BPD). Details in Appendix C.

