# OpenReview forum: "Likelihood Training of Cascaded Diffusion Models via Hierarchical Volume-preserving Maps"
_ICLR.cc/2024/Conference — ICLR 2024 spotlight_

### Official Review · Reviewer_hZLh · 2023-10-27

**Soundness:** 2 fair
**Presentation:** 3 good
**Contribution:** 3 good
**Rating:** 8
**Confidence:** 3

**Summary:**

The paper proposes the use of cascaded diffusion models for optimizing the likelihood of samples in diffusion models, with the motivation that cascaded models have shown excellent performance on image quality but have not been utilized as likelihood models. The paper goes on to present a method to facilitate likelihood modelling with cascaded diffusion by introducing *hierarchical volume-preserving maps*, allowing to express the model likelihood as a joint likelihood over different hierarchical scales. In practice, this includes wavelet and Laplacian pyramid transforms. The paper also proposes a surprising connection with the denoising score matching loss over the different scales of the hierarchical maps and optimal transport. The new models show impressive performance in model likelihoods on CIFAR-10 & different-resolution ImageNet datasets, out-of-distribution detection and lossless compression.

**Strengths:**

+ The paper clearly demonstrates that the proposed method provides improvements in log-likelihoods over non-cascaded models. As far as I am aware, the paper is the first to showcase that cascaded diffusion models can be used to improve log-likelihoods (in contrast to just the image quality).
+ The paper is well-written and easy to follow.
+ The new method is evaluated thoroughly, showcasing the performance of the model on applications relevant to log-likelihoods: OOD-detection and data compression.
+ I found the proposed connections to optimal transport particularly surprising, and this connection may be useful in further work.

**Weaknesses:**

## Minor issues
- The notation in some parts could be improved: E.g., eq. 27 takes the expectation over $\epsilon$ but refers to the noisy data variables $\tilde z^{(s)}$. On the next line, $\tilde z^{(s)}$ transforms to $\tilde x_k$. Small issues like this make the paper slightly difficult to read at times.

For other issues, see the questions part.

**Questions:**

### On the connection to the EMD metric:
- Given that it is the central part of the proof, I would like more elaboration on how exactly the connection to Theorem 2 of Shirdhonkar & Jacobs is made. One way to do this would be to repeat the theorem and its assumptions in the paper and showing step-by-step how does it apply here. In particular, I wasn’t able to see how do the $2^{-j(s+n/2)}$ terms in their statement of the Theorem connect to the application of it in the paper.
- What exactly does the paper claim by making the connection with score matching on wavelet representations and EMD? The last sentence of section 5.2. seems to claim that the method allows training diffusion models with the EMD measure, which does not quite seem to be the case here. While the connection between wavelets and EMD is definitely interesting, the connection to diffusion models doesn't seem particularly important as of now.
### On the ELBO and the necessity of volume-preserving maps:
- I might be confused, but it seems to me that volume-preserving maps are not necessary to form an ELBO with cascaded diffusion models. My thinking is as follows: We have the data $x$, and S sequences of latent variables $z_{1:T}^s$ with different resolutions and noise levels. The generative process, as defined in the paper, is $p_\theta(x,z)=p(x|z_1^S)\prod_{s=1}^S\prod_{t=2}^T p_\theta(z_{t-1}^s|z_{t}^s,z_1^{(<s)})p(z_T^s)$. We can then form the following inference process that is factorized for the different resolutions: $q(z|x) = \prod_{s=1}^S \prod_{t=2}^T q(z_t^s|z_{t-1}^s)q(z_1^s|x)$. Here $q(z_1^s|x)$ downsamples $x$ and adds the smallest level of noise in the diffusion process. Now if we form the ELBO, we get: $E_q[-\log p_\theta(x)]\leq E_q[-\log\frac{p_\theta(x,z)}{q(z|x)}]=E_q[-\log \frac{p_\theta(x|z_1^S)\prod_{s=1}^S\prod_{t=2}^Tp_\theta(z_{t-1}^s|z_t^s,z_1^{(<s)]})p(z_T^s)}{\prod_{s=1}^S \prod_{t=2}^T q(z_t^s|z_{t-1}^s)q(z_1^s|x)}]$, and further $=E_q[-\log p_\theta(x|z_1^S) - \sum_{s=1}^S[\sum_{t=2}^T\log\frac{p_\theta(z_{t-1}^s|z_t^s,z_t^{(<s)})}{q(z_t^s|z_{t-1}^s)} + \log p(z_T^s) - \log q(z_1^s|x) ]]$. From here, we can follow the standard derivation to get to the KL divergences (e.g., Appendix A in Ho et al., Eq.19., with the difference that their $x_{1:T}$ is redefined to $z_{1:T}^s$ and $x_0$ to $x$). I think this results in the same ELBO as in the submission, with the difference that instead of having a $p(x|z)$ term for each scale, this would only have it for the last scale. Do the authors agree with this point of view, or have I potentially misunderstood something? In case I have not misunderstood, this seems to be a major issue with the paper, since there is no comparison to cascading diffusion models without volume-preserving maps, making the significance of volume-preserving maps unclear.

## Overall

While the paper is well-written, the method showcases improvements and is well-evaluated, I hesitate to give an accepting score before I can see more clearly what is the benefit of volume-preserving maps. It is possible that I have missed something, and if so, I am willing to raise my score. Otherwise, I think that the paper requires more elaboration on what exactly is the role of volume-preserving maps in likelihood training of cascaded diffusion models. I would also like to see a clearer derivation for the EMD metric connection, as well as more elaboration on what is the significance of the connection.

References:
Ho et al., Denoising Diffusion Probabilistic Models, NeurIPS 2020

---

> ### Author Response · Authors · 2023-11-20
> **Reply to Reviewer hZLh**
>
> We greatly appreciate that the reviewer's close reading of our work, and their thoughtful concerns. Below, we address their reservations on the derivation of Theorem 5.1 and the necessity of the volume preserving map.
>
> **Eq. 27 -> Eq. 28.** The shift from $\mathbf{z}$ to $\mathbf{x}$ is correct and not a typographical error — this is due to Theorem 2 in Shirdhonkar. We provide a more in-depth treatment of this theorem for the reviewer, see below for details.
>
> **On the connection to the EMD metric.**
>
> *More detailed treatment of Theorem 2 in Shirdhonkar et. al.* We have provided two additional statements in Appendix A. Namely, Lemma A.1, which restates Theorem 2 in Shirdhonkar et. al according to our notation, and Lemma A.2, which shows in detail how Lemma A.1 can be applied to Theorem 5.1. To answer the reviewer’s immediate question, observe that for all possible values of $j, s, n$, the term $2^{-j(s + n/2)}$ is upper bounded by 1. Therefore it can be safely ignored in the upper bound, which is all we require for our Theorem 5.1 (Shirdhonkar et. al, 2008 provides both upper and lower bounds on the EMD).
>
> *Significance of Theorem 5.1 --- it does not seem like this allows diffusion models to be trained with the EMD measure.* Actually, this theorem shows precisely that diffusion models can be trained with EMD. Namely, by training with the loss function Eq. 17, our model minimizes the EMD between the true and predicted scores of the data distribution $\mathcal{X}$, which can be understood as unnormalized histograms in Eq. 19. This is the surprising result of Theorem 5.1 – a model that minimizes Eq. 17 is simultaneously a hierarchical model on $\mathbf{z}^{(s)}$ and an EMD-based model on the original $\mathbf{x}$. We believe that this connection is significant, since the EMD measure is known to have desirable properties for inferring distances between spatially structured data.
>
> **On the ELBO and the necessity of volume-preserving maps.**
> Indeed, when the goal is to compute a lower bound on the likelihood (i.e., the ELBO), the reviewer is correct, and volume-preserving maps are not strictly necessary. However, this derivation is different (and we argue, inferior) to our framework (Eq. 15) in two ways:
>
> First, the resulting ELBO is looser than our ELBO (Eq. 15), since it does not utilize the volume preservation property to obtain the equality in Eq. 9. Empirically, this difference is significant: in Appendix C and Tables 4-6 we compare our model to a hierarchical model that does not utilize a volume-preserving $h$ (i.e., a standard cascading hierarchy) derived without Eq. 9. With all other hyperparameters (architecture, noise schedule, and code implementation) held constant, our model improves significantly over the standard hierarchical baseline.
>
> Second, its inability to produce exact likelihoods $p_\theta(x)$ (e.g. Eq. 6) greatly limits its applicability. For example, the derivation does not work with autoregressive models or normalizing flows, while our framework does. Furthermore, a key property of diffusion models is that they can be interpreted both as an ODE (Eq. 6) and a latent variable model (Eqs. 3-5). As such, this derivation would also be useful for diffusion models that use the exact likelihood formula under Eq. 6 (such as Song et. al, 2020b, Song et. al, 2021, Lipman et. al, 2022, or Zheng et. al, 2023). While we derive our hierarchical diffusion model in its latent variable form, we note that all reported likelihoods (i.e., Tables 1-3) can also be computed exactly via Eq. 6.

---

> > ### Comment · Reviewer_hZLh · 2023-11-21
> > **Temporary response to rebuttal**
> >
> > I thank the authors for the new explanations and results! I will try to go through the EMD metric matter shortly, and then reconsider the score. One other question that came to mind: In Appendix C, do you evaluate LP-PCDM and W-PCDM models with the ELBO or the exact likelihood evaluation using the ODE? In case it is the latter, the ELBO evaluations would seem useful to include as well, so that we can assess whether the improvement is due to the ability for exact likelihood evaluation or due to the ELBOs themselves being better as well.

---

> > > ### Author Response · Authors · 2023-11-21
> > >
> > > Definitely, please take your time. To answer your other question, we actually do use the ELBO to evaluate the LP-PCDM and W-PCDM models rather than the ODE in Eq. 6, which indeed suggests that the ELBOs themselves are better in this case.

---

> > > > ### Comment · Reviewer_hZLh · 2023-11-22
> > > > **Response to rebuttal**
> > > >
> > > > I now went through the new derivations regarding the EMD, nicely written! I am satisfied with all the answers, and indeed I missed the point about exact likelihoods on the first reading. I think that this is good work, and will raise the score accordingly.

---

### Official Review · Reviewer_y69M · 2023-10-29

**Soundness:** 3 good
**Presentation:** 2 fair
**Contribution:** 3 good
**Rating:** 8
**Confidence:** 5

**Summary:**

This paper shows that volume-preserving maps can be used to define multiscale (cascaded) generative models with tractable likelihood. Extensive numerical experiments demonstrate the improvements in likelihood modeling, out-of-distribution detection, and compression achieved by the ability to use multiscale models. The paper also makes a connection between multiscale diffusion models and optimal transport with the earth's mover distance.

**Strengths:**

The authors tackle an important problem, which is the use of cascaded approaches for likelihood modeling. The proposed solution is simple (which I mean in the best possible way!) and the numerical experiments are solid and convincing. The paper is also clearly written (except that the core concept is not stated until page 6, see Weaknesses).

**Weaknesses:**

- The theoretical setting of the paper is general, with possibly non-linear volume-preserving maps $h$, but in practice only linear $h$ are used, which thus boil down to using an orthogonal transform (for wavelets) or a tight frame (for Laplacian pyramid). The linear setting already leads to important improvements, and the possibility of the extension to non-linear maps is interesting for future research, but it obfuscates slightly the results of the paper behind unnecessary complexity. In particular, using "Hierarchical volume-preserving maps" in the title instead of a more explicit reference to orthogonal wavelet transforms (which yield the best results) does not accurately describe the contents of the paper and makes it harder for the reader to connect the contributions of the paper to known concepts. Orthogonal transforms are not mentioned anywhere before page 6, yet are the core component of the method.
- Related to the previous point, orthogonal wavelets have been used for multiscale generative modeling in several missed previous works: see [1] for nomalizing flows, [2] for direct likelihood modeling, and [3] for diffusion models (closest to the setting of the paper, though the focus is on image quality rather than likelihood modeling). In particular, [1] and [2] critically rely on the orthogonality of the transform to model the likelihood. Also, the recombination of $z^{(<s)}$ into a same-resolution image mentioned in Appendix B is crucially used by [1-3] to parameterize efficiently the conditional likelihood.
- The connection to the earth's mover distance seems problematic to me. First, it is unclear what benefits it brings to the paper: I do not understand how to interpret Theorem 5.1, since here the transport is between the true and approximated scores (seen as distributions on $\mathbb R^2$), as opposed to transport between distributions of images as is usually the case. Second, the soundness of the results look questionable to me. For instance, the Wasserstein-p distance is defined between probability distributions, which thus requires that the scores have non-negative entries that sum to one. The authors do not mention this restriction anywhere in the paper. Second, the proof seems to assume that the wavelet coefficients of the image score $\nabla \log q(x)$ are the wavelet conditional scores $\nabla \log q(z^{(s)} | z^{(<s)})$, but this is not the case. Indeed, one has $\log q(x) = \log q(z^{(1)}) + \log q(z^{(2)} | z^{(1)})$ (with $S = 2$ for simplicity), and thus the $z^{(1)}$ component of the image score includes an additional term coming from the high-frequency score component (called "free energy" in [2]). For this reason, the equivalence between wavelet and Laplacian pyramid for the score matching losses also breaks down (which is a more plausible explanation for the difference in performance than the authors' footnote 1).

Despite these weaknesses, I recommend acceptance (see Strengths above). I am willing to increase my score if the authors address these points.

[1] Jason J Yu, Konstantinos G Derpanis, and Marcus A Brubaker. “Wavelet Flow: Fast Training of High Resolution Normalizing Flows.” In Advances in Neural Information Processing Systems, 2020.

[2] Tanguy Marchand, Misaki Ozawa, Giulio Biroli, and Stephane Mallat. “Wavelet Conditional Renormalization Group.”

[3] Florentin Guth, Simon Coste, Valentin De Bortoli, and Stephane Mallat. "Wavelet score-based generative modeling." Advances in Neural Information Processing Systems, 2022.

**Questions:**

- If $z$ is higher-dimensional than $x$, then $z = h(x)$ does not admit a probability density (it is not absolutely continuous with respect to the Lebesgue measure). How do the authors deal with this difficulty, and how is Lemma 4.1 not meaningless in this case?
- The authors mention switching to a different architecture and weighting scheme for optimizing FID as opposed to likelihood. Does the improvement in FID come with an increase in BPD? If so, the tradeoff between the two should be acknowledged explicitly.
- Why not using $M=1$ in the OOD detection task? In high-dimensions, we expect from concentration of measure that $-\log p(x) \approx \mathbb H(p(x))$ for almost all $x$, so we should be able to detect changes in distribution from just one sample?

Minor suggestions:
- Multiscale image modeling has a much older history than 2017, e.g., [4-10] for a very incomplete list, which seems more relevant than the reference [Horstemeyer, 2010]. Though these earlier works generally focus on other tasks such as denoising or compression, the motivations for using a multiscale representation remain the same.
- Line before equation (5) should read $\log p_\theta(x_0)$ instead of $p_\theta(x_0)$.
- Any linear map could be used for cascaded likelihood modeling, even if its determinant is not one (as long as it is not zero). Indeed, it just introduces a constant offset in the likelihood, which has no effect for training, and this offset can be estimated once offline for test-time likelihood evaluation. In particular, the standard cascaded hierarchy is in this case.
- The discussion below Definition 1 should state that it requires $\mathrm{dim} \mathcal{Z} \geq \mathrm{dim} \mathcal{X}$ with a full-rank Jacobian $\mathrm{rank}(A) = \mathrm{dim} \mathcal{X}$. Also, the convention used for the Jacobian matrix should be stated to avoid confusions with its transpose: i.e., $A \in \mathbb R^{\mathrm{dim} \mathcal{Z} \times \mathrm{dim} \mathcal{X}}$.


[4] P J Burt and E H Adelson. The Laplacian pyramid as a compact image code. IEEE Trans Comm, Apr 1983.

[5] S Mallat. A wavelet tour of signal processing: The sparse way. Academic Press, 2008.

[6] A Chambolle, R A DeVore, N Lee, and B J Lucier. Nonlinear wavelet image processing: Variational problems, compression, and noise removal through wavelet shrinkage. IEEE Trans Image Processing, Mar 1998.

[7] R W Buccigrossi and E P Simoncelli. Image compression via joint statistical characterization in the wavelet domain. IEEE Trans Image Processing, Dec 1999.

[8] M J Wainwright, E P Simoncelli, and A S Willsky. Random cascades on wavelet trees and their use in modeling and analyzing natural imagery. Applied and Computational Harmonic Analysis, Jul 2001.

[9] L Şendur and I W Selesnick. Bivariate shrinkage functions for wavelet-based denoising exploiting interscale dependency. IEEE Trans Signal Processing, Nov 2002.

[10] J Portilla, V Strela, M J Wainwright, and E P Simoncelli. Image denoising using scale mixtures of Gaussians in the wavelet domain. IEEE Trans Image Processing, Nov 2003.

---

> ### Author Response · Authors · 2023-11-20
> **Reply to Reviewer y69M**
>
> We greatly appreciate the time and care the reviewer took to study our work, the in-depth discussion of our theoretical contributions, and the encouraging comments. Below are detailed responses to their concerns.
>
> **The framework proposed considers all homeomorphisms that satisfy Eq. 8, but only two linear examples of $h$ are empirically evaluated.** We believe a major contribution of our work is demonstrating general conditions under which hierarchical models also yield tractable and exact likelihood models, which can greatly inform future research on diffusion models, many of which are hierarchical. We leave further exploration of various $h$, including non-linear cases, to further work. We do recognize that the linear scope of our experiments should be clearly stated early on, and we have added sentences clarifying this point to the abstract, introduction, and Section 4 (introducing the hierarchical volume-preserving map). To quote our added statement in the list of contributions in Section 1:
> “We show that two well-known multiscale decompositions, the Laplacian pyramid and wavelet transforms, are hierarchical volume preserving maps, where the linearity of the maps reduce the volume-preserving property into more familiar notions of orthogonality and tight frames (Unser et. al, 2011).”
>
> **Connection to the EMD Metric.**
>
> *Transport between scores, rather than images.* Indeed, Theorem 5.1 relates our training loss (Eq. 17) to the transport between scores of the images, rather than images themselves. However, this is not so out of the ordinary. The notion of applying Earth Mover’s Distance to features of the image histogram, rather than its pixels directly, is a relatively common strategy in the literature — for example, the seminal work Rubner et. al, 2000 (as cited in our text) finds that the EMD distance on image “signatures” such as texture, color, and joint color/position embeddings work better than the raw pixels themselves.
>
> *The Wasserstein-p distance is defined between probability distributions, and the score is not a probability distribution.* In Eq. 18, we do in fact consider the setting where $x$ and $y$ are unnormalized and do not sum to one (see the sentence leading up to Eq. 18). This unnormalized case is known as partial matching, and is a setting considered in Rubner et. al, 2000 as well as Shirdhonkar et. al, 2008 which our theory builds off of (see respective references in our work). On the other hand, non-negative entries in the score can indeed be problematic, but this can be circumvented by subtracting the most negative entry from both $x$ and $y$. Namely, let $c$ be such that $c \leq x_{ij}$ and $c \leq y_{ij}$ for all $i, j$. Then $x - c$ and $y - c$ are surely non-negative. Moreover, inspecting Eq. 18, we can see that this transformation does not affect the unnormalized Wasserstein metric, since the $c$ from each unnormalized histogram cancels.
>
> *Free energy term.* We do not quite follow the reviewer’s argument here. What is the reviewer referring to as the “additional free energy term” included in the $z^{(1)}$ component of the wavelet score? As it stands, the equation provided by the reviewer appears to agree entirely with our original formulation (Eq. 15).
>
> **Preserving likelihoods when $z$ and $x$ are not the same dimension.** We agree that likelihood computation is much trickier when $z$ and $x$ are not the same dimension. However, we can preserve a pushforward measure when we assume $z \in \mathcal{M}$ and $x \in \mathcal{N}$ are from differentiable manifolds $\mathcal{M}$ and $\mathcal{N}$, and $h$ is a diffeomorphism between the two spaces (we recommend referring to [1, 2]). This is satisfied because $h$ is linear and invertible in the cases we consider.
> [1] Foundations of differential geometry. S Kobayashi, K Nomizu. 1996.
> [2] Normalizing flows on riemannian manifolds. M Gemici, D Rezende, S Mohamed. 2015.
>
> **Acknowledging the trade-off of a likelihood weighting scheme optimizing FID.** Optimizing for FID, we obtain BPD 9.55 and 10.31 respectively for W-PCDM and LP-PCDM, respectively. We have added an acknowledgement of this trade-off.
>
> **Why not use $M=1$ for OOD detection?** Thank you for this suggestion! We tried using $M=1$, and saw little appreciable change in performance (AUROC of out-of-detection classification of SVHN images was reduced by a hundredth of a point), so we have included this result in our work instead of the $M=2$ case, which we had chosen following the experiments in Nalisnick et. al, 2019. Anomaly detection with $M=1$ is clearly more difficult than $M=2$, so this is a better result. As the reviewer suggests, the superior performance even at $M=1$ may be closely related to the concentration of measure in high dimensions.
>
> **Minor suggestions.** We thank the reviewer for bringing these errors to our attention, and have incorporated them into the text.

---

> > ### Comment · Reviewer_y69M · 2023-11-21
> >
> > - Thank you for your precisions on the interpretation of the Wasserstein distance on unnormalized possibly negative images. I realize I made a mistake in my review: I confused the "forward" score $\nabla \log q(x_k|x_0)$ with the "backward" score $\nabla \log q(x_0|x_k)$.
> > - I agree that a probability distribution can be defined even without a density. But since the goal of the paper is to address likelihood modeling, and thus computes (logs of) probability densities, I insist it seems problematic to me to model the likelihood of a variable $z = W^T x$ which does not admit a density because it is supported on a lower-dimensional space (the range of $W^T$).
> >
> > As a result of the discussion, I am increasing my score to 8.

---

### Official Review · Reviewer_ocEJ · 2023-11-06

**Soundness:** 2 fair
**Presentation:** 2 fair
**Contribution:** 3 good
**Rating:** 8
**Confidence:** 2

**Summary:**

This paper proposes considers multi-scale diffusion models, and shows theoretically that one needs volume-preserving scale transformations to make the likelihood behave “nicely”. The empirical results are excellent.

---

Post-discussion update: My concerns were sufficiently addressed, and all the confusions I had were clarified. This is a high-quality, substantial contribution to diffusion models from solid theoretical understanding.

**Strengths:**

- Excellent empirical results, which are great throughout.
- The method sheds light on the Laplacian and Wavelet decompositions, and proposes a relatively principled training scheme for them. The theoretical analyses of the scale transformations are delightful to see, and this discussion is a significant viewpoint to diffusion models

**Weaknesses:**

- The theoretical presentation is informal and imprecise. I couldn’t follow the theory or substantiate the claims made by the paper. The final model is also ultimately undefined and feels very adhoc despite the principled approach. It’s also confusing what is the contribution of this paper, since it seems that ultimately the paper just uses the Laplace/Wavelet transforms as-is, which are already well known.
- The experiments do not compare to recent diffusion works [there is no comparisons to 2023 methods, and only one 2022 method (!)], and lack direct comparisons to diffusion models with super-resolution, wavelet or laplace transforms. The experiments are quite roundabout and describe just the overall performance, and don’t directly substantiate specific contributions made in this paper. It is then difficult to see why we improve, or if the theoretical contributions had anything to do with it. There are no ablations. There are very little result exposition or illustrations.

**Questions:**

Minor comments

- The eq 3 is looks like the simple forward process, but is instead the forward “posterior”. For instance, DDPM paper (Ho et al 2020) eq 2 LHS is the same equation as eq 3 LHS here, but has a conflicting RHS. Can you clarify which q interpretation you are using, and which q the eq 5 is taken over.
- I’m not sure I if I understand the premise of the paper. I agree that with invariant transformation we have p(x) = p(h(x)) property. However, this is useful only if the joint distribution p(z^1, …, z^S) decomposes independently into p(z^1) * … * p(z^S); and I don’t see why this would happen. Surely the different scales are highly dependent, and you would have to do a chain rule p(z^S | z^S-1) … p(z^2 | z^1) p(z^1) [or some other decomposition], and in this setting I don’t see the invariant property being useful. The paper should clarify the invariance wrt scale independence, preferably in the introduction and in beginning of sec 4. Right now the main claim of the paper does not convince. For instance, *“if h(x) is the scales of the hierarchical model, then we can directly use the joint likelihood over these scales as the desired model likelihood”* is generally false as far as I can see.
- Eq 8 has square root, but why? It does not seem to do anything when det=1.
- What does z(1), z(2), . . . , z(S) = h(x) mean? I don’t understand the notation. Does h(x) return S different things, or does h return one thing as in z1=h; z2=h; etc? h seems to be a downscaling operator, is this true? The nature of “h” is a bit confusing at this stage of the paper. What does h output? What resolution?
- What does “p_theta trained on z1..zS” mean? Trained how? Against what objective? What kind of p_theta? These statements are overly casual, and need to be made precise, explicit, rigorous and transparent.
- “Then the model likelihood”. What does model likelihood mean? Can you make this rigorous? This is too informal. Or is this supposed to be informal and imprecise?
- What does p_theta(x) mean in eq 9? Does it refer to eq 7 or something else? What does it mean to apply h(x) to eq 7: do we apply h() to all z’s together, or separately, or what..?
- I don’t think the statements after eq 9 are true; I see no convincing arguments for this. It’s not clear what the terms in eq 9 even mean or how they are defined, and the dependency issue of the joint still remains.
- In eq 10 the h is now some kind of linear operator, while earlier it was a function. Is the idea that h(x)=h*x? It would be good to clarify the distinction between function h and operator tensor h.
- What does * mean in (j*2)? Is this a product or convolution? If this is a convolution, what does it mean to convolve j with 2?
- What does fig 2 show? I think it shows y^4 and z^3...z^1. Can you clarify?
- What does tight frame mean? Can you give a conceptual explanation? What does “h” mean in parseval equation? There is no “h” in the Laplace section of the paper at all: does some of this stuff (y,z,d,u) relate to h?
- Again, what is the “mapping” h in Wavelet case? Is it the z’s, or y’s, or them together, or something else?
- The wavelet and Laplacian pyramids both retain the original resolution of the image. Wasn’t the goal of this paper to reduce resolution? I’m a bit confused. Can you clarify the notion of resolution vs scale, and clarify which one you want to reduce? The very first sentence of the paper in introduction talks about super-resolution, which leads me to believe that one should change resolution somewhere in the paper, but neither of the transform types seems to change resolution. Can you clarify these aspects?
- I don’t see what lemma 4.1. has to do with eq 15. It seems that you just apply the x=z1…zS substitution and use chain rule to split the p(x) to p(zs|z_<s). Where is lemma 4.1 here? Also, the paper has already defined p(x) in eq 7, which seems to conflict with the p(x) in eq 15. Can you clarify? Is this a redefinition, or are both true?
- What is the q in eq 16?
- What is z_0^s in eq 16? How do you get them? How do you get the z_1’s? How do you get the z_k’s? What are the terms here: I can’t really follow any of them. Can you explain all three terms (the logp and two KLs). For instance, the regular logp term is the probability of observed image given z^1 while marginalising all intermediate states away. Here isntead we have p(z^s_0 | z_1^s, z_0^<s). So we are looking at one scale likelihood, but somehow conditioning with earlier scale “observations”? I can’t really follow at all what is happening, since very little of eq 16 or the underlying processes have been defined or characterised. Please include an algorithm boxes for training and sampling. It’s also confusing what is h here.
- I’m having hard time understanding what the eq 18 is conceptually representing. Is the idea that criss-cross the pixels from one image to another by moving the pixel locations around?
- Eq 19 is a lot of stuff without much motivation, introduction or exposition. I have hard time understanding what this means or how it connects with the paper. Somehow optimising the scale losses is equivalent to reordering pixel locations in the scores…? Err.. what? This feels very strange, and I can’t follow. I also don’t see why this is significant: what are you trying to argue here? How do we benefit from this connection? If you want to present this theoretical connection, it needs to be presented in a way that is digestable, and requires more exposition and explanation, and also helpful illustrations.
- At experiments I still don’t know what does the function “h” mean in this paper, or what it is. I guess it’s the Laplace pyramid, but not sure which part. I’m also not sure how did we now solve the problem of joint needing to be marginalised to only evaluate the final image. I can’t really connect the problem statement in introduction to the methods presented in the paper. I think the paper needs a method summary section that explains how all the pieces come together to solve the original problem, while including an algorithm box.
- The paper should cite to simple diffusion, that also tackles multi-resolution.
- In experiment the key comparison is how does this method fare against diffusion models that utilise laplace or wavelet pyramids; or have an explicit super-resolution component. Looking at table 1 almost all comparison targets are irrelevant old methods, and I fail to see many if any super-resolution/laplace/wavelet methods. There is only a single comparison to a 2022 method, and none to 2023. Given the astronomical pace of the research in diffusion, this is not acceptable. The paper needs to compare both to (i) other super/wavelet/laplace diffusion models, and more comprehensively to 2022 and 2023 diffusion models. The claims of state-of-the-art performance is unsubstantiated.
- The OOD experiments only seem to compare to quite old generative models. Why not compare to other diffusion models? Wasn’t the point of the paper to improve the multi-scale handling of diffusion models, so surely one should then compare to other diffusion models with more adhoc multi-scalings, or to diffusion models with a single scale only. I’m also not sure what the table 2 rows mean.
- In 6.3. there are comparisons to uni-scale diffusion models (which is great), but one should also compare to competing multi-scale diffusion models.

---

> ### Author Response · Authors · 2023-11-20
> **Reply to Reviewer ocEj (1/3)**
>
> We appreciate the reviewer's thoughtful comments and insightful critiques. We also thank the reviewer for their extensive questions. We acknowledge that our work builds extensively on existing principles in computer vision, statistics, and signal processing, and hope that our point-by-point response has provided some intuition for the principles underlying our work.
>
> **Informal and imprecise theoretical presentation.**
> We do take very seriously criticisms of the informality and incorrectness of our text. If the reviewer’s concerns with the theoretical presentation are related to the other points (and minor comments) further in the review, we hope we have addressed them in a satisfactory manner and the reviewer may reconsider this judgment. We are happy to answer any further questions.
>
> **Contribution of the paper.**
> The paper uses Laplacian pyramid / Wavelet transforms as is. To summarize our contributions, we have proposed a generalized framework for hierarchical modeling that allows for tractable likelihood computations, which was previously not considered or possible with related works. While two notable maps are the Laplacian pyramid / Wavelet transform, our framework is certainly not limited to these transforms. We are not aware of Laplacian pyramid / wavelet diffusion models that derive tractable likelihoods or even conduct density estimation experiments. Finally, we demonstrate significant improvements to the state-of-the-art with our hierarchical framework, and enable future work with more complex hierarchical volume-preserving maps.
>
> **There are few / no comparisons to 2022 and 2023 methods.**
> We did not include more recent results as we did not find significant improvement over existing results in Table 1. As per the reviewer’s request, we have incorporated several more 2022 and 2023 methods (Soft Truncation Kim et. al 2022, INDM Kim et. al 2022, i-DODE Zheng et. al 2023). Crucially, we still obtain significant improvements over these methods. We would be happy to include any other results that the reviewer suggests.
>
> **There are no comparisons to diffusion models with super-resolution, wavelet, or laplace transforms.**
> As discussed in our work, there did not previously exist super-resolution, wavelet, or Laplacian pyramid -based diffusion models that preserve likelihood computation. Therefore, we cannot form any comparisons on the experiments in Section 6. That being said, as pointed out by another reviewer, it is possible to fold the latent scales of standard cascading hierarchies (that do not satisfy Definition 1) into a looser variational lower bound. However, this comes at a significant cost to modeling performance, as seen in Table 4-6 (Appendix C).
>
> **The experiments are roundabout, and don’t directly substantiate specific contributions made in this paper.** Can the reviewer elaborate on this point? If it is due to points in the minor comments below, we hope we have adequately addressed this criticism. The two central claims in this paper are that 1) we enable exact likelihood estimation, which is substantiated by Lemma 4.1 and 2) that hierarchical modeling with wavelets and Laplacian pyramids approximate an EMD loss on the scores, which improves modeling performance and is substantiated by significant improvements in performance (Tables 1-3).
>
> **There are no ablations.** Our paper claims that hierarchical volume-preserving maps allow for likelihood modeling in cascaded models. Unfortunately, hierarchical maps that are not volume-preserving (such as a standard cascading model) do not allow for exact likelihood computation, therefore direct ablations are not possible. However, as previously mentioned, one can still obtain a loose loser bound on the likelihood. We compare our work against this lower bound in a series of ablations in Appendix C and Tables 4-6, and find that using a standard cascading hierarchy (end of Section 4.1) significantly reduces modeling performance.
>
> **Minor comments:**
>
> **Eq. 3 differs from Ho et. al, 2020.** The RHS here is different from Ho et al, 2020 because they are different Markov processes. Eq. 2 in Ho et. al, 2020 describes the forward diffusion process, whereas Eq. 3 here describes the reverse diffusion process. The $q$ in Eq. 5 is also taken over the same reverse diffusion. For more information, see Appendix E.1 of Kingma et. al, 2021 (as cited in our work).

---

> > ### Comment · Reviewer_ocEJ · 2023-11-21
> > **resp**
> >
> > Thanks for the clarifications. I still have two major concerns.
> >
> > **On math**. I don't understand the math on the likelihood preservation. So in eq 7 we define that $x$ becomes a collection of scales $h(x) = (z^1, \ldots, z^S)$, and we have a well-defined marginal likelihood in eq 7. Then in following paragraph we state that $p(x) = p(h(x))$. Ok, but what does this actually do? How does this equation expand? How does the $h$ go inside eq 7? I think this then means that we have $p(x) = p(z^1,\ldots, z^S)$ without the integrals, so this would be an alternative (and possibly conflicting) definition to eq 7. We now have two definitions for $p(x)$. Eq 7 says that probability of one image necessitates integrating out infinitely many latent scales; and yet this new definition $p(x) = p(h(x))$ say that we don't need to do that. Can you explain the intuition here?
> >
> > I'm also confused what does $p(z^1, \ldots, z^S)$ mean. How can the density $p_\theta$ admit both $x$ and $z^1,\ldots,z^S$ (and also later $z^1$ alone, and $z^s$ conditioned on $z^{<s}$ in eq 15)? This density seems overloaded: what are the definitions for all these $p_\theta$'s, and are they the same density or different? How are they defined?
> >
> > I'm also confused where does eq 15 come from. How can you define this? Given that $p_\theta$ is likely a neural network, surely it will not give you same output for $x$ and $z^1,\ldots,z^S$. This also conceptually makes little sense to me. First we have the probability of lowest scale $p(z^1)$, and then we add to it a bunch of higher-scale transition probabilities $p(z^s|z^{<s})$ (or whatever this means). This means that we add together a bunch of scalar density values, resulting in one density value. Why would this value somehow match the density value $p(x)$ of the original image?
> >
> > If the volume is preserved, shouldn't then all $p(z^s | z^{<s})$ be equal as well? Why are these allowed to be different? And if they are allowed to be different, then don't we violate the volume-preservation?
> >
> > Intuitively for me the volume-preserving map only makes sense as a single step: it would make sense to say that $z^s = h(z^{s-+1}) and then state that $p(z^{s-1})$ equals $p(h(z^s)) = p(z^{s-1})$. Yet, no equation like this is in the paper.
> >
> > I would like to see a convincing derivation that shows concretely how $p(x) = p(h(x))$ is true where you rigorously define the used $p$ functions as well.
> >
> > As you can see, I'm struggling to understand. Can you clarify?
> >
> > **On claims**. I understand the paper contributing an improved loss for "multi-scale" settings, while earlier diffusion methods used more adhoc losses in their approaches to multiscale/superresolution/cascades/etc. The paper then should provide evidence how does the new loss improve over the earlier diffusion models that were less rigorous, irrespective of whether their likelihood was tractable or exact.

---

> ### Author Response · Authors · 2023-11-20
> **Reply to Reviewer ocEj (2/3)**
>
> **Premise of the paper.** We do not understand the reviewer’s concern here. Why must we assume independence of the scales $z^{(1)}, z^{(1)}, \dots, z^{(S)}$ for the invariance to hold? In fact, we consider solely the dependent case, as this is the necessary conditional structure for a hierarchical multiscale model.
>
> **Square root in Eq. 8.** Indeed, the square root is not strictly necessary here. However, this entire term (including the square root) is mathematically meaningful in normalizing flows and differential geometry as the instantaneous change in volume between differentials (infinitesimal volumes) of the two Riemannian manifolds at $x$ and $h(x)$ (Berger and Gostiaux, 2012), so we have opted to keep the square root.
>
> **What does $z^{(1)}, \dots, z^{(S)} = h(x)$** mean? Since $h$ defines a hierarchical map, the range of $h$ is the cartesian product of the $S$ scale spaces, i.e., $h: \mathcal{X} \rightarrow \mathcal{Z}^{(1)} \times \mathcal{Z}^{(2)} \times \dots \times \mathcal{Z}^{(S)}$, where $x \in \mathcal{X}$ and $z^{(s)} \in \mathcal{Z}^{(s)}$. We have swapped sides of the equality and added parentheses around the RHS in the equation $h(\mathbf{x}) = (z^{(1)}, \dots, z^{(S)})$ to emphasize that the functional output of $h$ is a cartesian product, and further clarified the functional definition of $h$ in Section 5.1.
>
> **How is $p_\theta$ “trained” in Lemma 4.1?** We have removed the word “trained” in this sentence. $p_\theta$ need not be trained in any manner — the theorem holds as long as $p_\theta$ is a valid likelihood function on $\mathbf{z}^{(1)}, \dots, \mathbf{z}^{(S)}$.
>
> **What is $p_\theta(x)$? What is the model likelihood?** The model likelihood is the joint likelihood of the model and data, rigorously defined as $p_\theta(x)$. $p_\theta$ simply refers to a likelihood function that takes a data point (or its latent representation) as input, and returns its modeled density. We have changed "model likelihood" to simply "likelihood" to reduce confusion.
>
> **The statements after Eq. 9 are not true.** We state after Eq. 9 that Eq. 9 (i.e., $\log p_\theta(x) = \log p_\theta(h(x))$) demonstrates a form of probabilistic invariance. Formally, letting $g_x(h) = p_\theta(h(x))$, the set of hierarchical volume-preserving maps $\mathcal{H}$ form an equivalence class under the the equivalence relation $g_x(h) = g_x(h’)$ for $h, h’ \in \mathcal{H}$ and $x \in \mathcal{X}$. This equivalence relation can be clearly seen by applying Eq. 9 to $h$ and $h’$, and invoking the transitive property.
>
> **$h$ in Eq. 10 is a linear operator. Isn't it a function?**
> Indeed, in Eq. 10, $h$ is a function that is happens to also a linear operator, since we are considering the standard cascading hierarchy as described in Ho et. al, 2020, which can be seen as a bilinear resampling operation. In general, $h$ can be an arbitrary function that satisfies Eq. 9.
>
> **The * in (j*2).** The asterisk is simply the product, not a convolution. We see how it can be confusing, and have replaced the asterisk with a dot (i.e., $j \cdot 2$) for clarity.
>
> **What does Figure 2 show?** In this figure, we show the Laplacian Pyramid coefficients, i.e., $z^{(1)}, z^{(2)}, …, z^{(4)}$ from left to right.
>
> **What is a tight frame?** Informally, a tight frame describes a representation (of a signal) that is norm-preserving. For a more formal treatment, please see Unser et. al, 2011 (as cited in our manuscript).
>
> **What is $h$ in the Laplacian Pyramid section in 4.2? What is $h$ in the Wavelet Decomposition section in 4.2?** $h$ refers to the Laplacian pyramid or wavelet transform, respectively. Formally, $h(\mathbf{x}) = (z^{(1)}, z^{(2)}, …, z^{(S)})$ where where $(z^{(1)}, z^{(2)}, …, z^{(S)})$ are as defined in the respective subsection (Wavelets or Laplacian Pyramids) in Section 4.2. While the functional form is not explicit to maintain generality w.r.t. $S$, this definition is rigorous.
>
> **Wasn’t the goal of this paper to reduce resolution? Resolution vs scale?**
> The goal of the paper is not to reduce resolution. Hierarchical models are generally useful because they enable an explicit multi-scale model of the data. While super-resolution is used to obtain finer scales from coarser scales (e.g. $\mathbf{z}^{(s)}$ from $\mathbf{z}^{(s - 1)}$), all scales are retained in our model, and therefore information is not lost. In fact, in Ho et. al, 2022 (in the paper) the sequence of scales actually contains more parameters (i.e., pixels) than the original image. Moreover, when $h$ is the wavelet transform, the size of the largest scale is 25% the original image.

---

> ### Author Response · Authors · 2023-11-20
> **Reply to Reviewer ocEj (3/3)**
>
> **Relation between Lemma 4.1 and Eq. 15.** The reviewer’s derivation is not true, as $x \neq z^{(1)}…z^{(S)}$. $x$ is only equivalent to its scales under the transformation $h$, i.e., $h(x) = z^{(1)}…z^{(S)}$. In general, transformations (like $h$) do not preserve likelihoods. In other words, $p_\theta(\mathbf{x}) \neq p_\theta(h(\mathbf{x}))$ for an arbitrary function $h$. Therefore, to our knowledge Eq. 15 is not true unless Lemma 4.1 holds and $h$ is a volume-preserving map.
>
> **What is the q in eq 16?**
> The $q$ in Eq. 16 is the same $q$ as in Eqs. 3 and 5.
>
> **What are the $z_{0:T}^{(s)}$ in Eq. 16?** $z_{0:T}^{(s)}$ are the variables of the diffusion process corresponding to each scale $s$, where $z_0^{(s)}$ = $z^{(s)}$. This is much like $x_{0:T}$ are variables of standard diffusion process as described in Section 3 where $x_0 = x$, except that the diffusion described by $z_{0:T}^{(s)}$ are additionally conditioned on the prior scales $z^{(<s)}_0 = z^{(<s)} = (z^{(1)},z^{(2)}, \dots, z^{(s-1)})$.
>
> **Understanding Eq. 18.** This equation defines a distance between two distributions $\mathbf{x}$ and $\mathbf{y}$ based on the cost to transport mass from each pair of locations ($(i,j)$ in $\mathbf{x}$ and $(k, \ell)$ in $\mathbf{y}$) so that $\mathbf{x}$ looks like $\mathbf{y}$. The infimum ensures that this cost is the most reasonable (i.e., efficient) transport plan. Intuitively, we are trying to measure the effort required to shift pixel intensities from the “high mass” areas of $\mathbf{x}$ towards the “high mass” areas of $\mathbf{y}$, until $\mathbf{x}$ resembles $\mathbf{y}$. For a rigorous understanding, we encourage the reader to look into more detailed treatments on optimal transport [1] or Earth Mover’s Distance (Rubner et. al, 2000).
>
> [1] Computational Optimal Transport. https://arxiv.org/abs/1803.00567
>
> **Understanding Eq. 19.** We agree with the reviewer that Theorem 5.1 is quite surprising! The main theoretical machinery for this result can be found in Theorem 2 of Shirdhonkar and Jacobs, 2008 (as cited in our manuscript). In essence, the multiscale decomposition offered by wavelet and laplacian pyramid transformations allow us to approximate Eq. 18 via differences between the corresponding scales of $\mathbf{x}$ and $\mathbf{y}$. This is significant for two reasons. First, Eq. 19 is much easier to compute than Eq. 18. Second, Eq. 19 is also known as the Earth Mover’s Distance, which is a well-known measure for distances between spatially structured data (e.g. images) that works better than the L2 distance. Therefore, Theorem 5.1 suggests that the performance of our algorithm could be directly attributed to the hierarchical prior.
>
> **What is $h$ in Section 6 (experiments)?** We discuss $h$ at length in Section 4. In general, $h$ is a hierarchical volume-preserving map — any function that maps data (e.g., images) $\mathbf{x}$ to a sequence of latent variables $\mathbf{z}^{(1)}, \dots, \mathbf{z}^{(S)}$ and satisfies Eq. 8. In our experiments, $h$ refers to either the Laplacian pyramid transform or the wavelet transform (LP-PCDM and W-PCDM, respectively).
>
> **Citing simple diffusion.** If the reviewer is referring to “simple diffusion: End-to-end diffusion for high resolution images” by Heek et. al, 2023, it appears that this method specifically does not involve any multiscale (i.e., hierarchical, cascading) architecture. (See e.g. Section 1 and Figure 2 caption of their paper.) Our understanding is that they tackle diffusion purely on the pixel space.
>
> **Why are there no comparisons to other multiscale diffusion models (Section 6.1, 6.2, 6.3)?** There are no other multiscale diffusion models that produce tractable likelihoods — in fact, this is the main contribution of our work. Tractable likelihoods are required for any experiments in Sections 6.1, 6.2, and 6.3. The only known method to obtain a lower bound on the likelihood is via a variational lower bound. We derive and compare against this setting in Appendix C and Tables 4-6, and show that our method performs significantly better. On the other hand, we have added more results for standard diffusion models 2022 and 2023. We note that we still demonstrate significant improvements over these results. We are happy to include any other recent works, or likelihood-based multiscale diffusion models the reviewer suggests.
>
> **Comparing OOD benchmark to modern diffusion models.** We have included a column in Table 2 against the state-of-the-art diffusion model in likelihood modeling, the variational diffusion model (VDM, Ho et. al, 2021).
>
> **What do the rows in Table 2 mean?** Why is there no comparison against modern diffusion models? We include detailed information about the experiment in Table 2 in Appendix B.1. We compare against the current state-of-the-art diffusion model, variational diffusion models (VDM, Ho et. al, 2021).

---

> ### Author Response · Authors · 2023-11-21
> **Reply to Reviewer ocEJ**
>
> Thank you for your response, we are more than happy to clarify.
>
> **On math.**
>
> *$p(x)$ has two definitions.* First, neither Eq. 7 nor Eq. 9 are definitions. Eq. 7 is obtained by marginalization, which is a fundamental property of expectations (https://en.wikipedia.org/wiki/Marginal_distribution). Eq. 9 is obtained via Lemma 4.1, which builds from the continuous change of variables formula (see, e.g., a simpler formulation in [1] or the full manifold-based formulation in Ben-Israel, 1999 in our text), once again derived from fundamental principles in probability theory. Intuitively, the key difference here is the assumptions on $h$. The marginalization formula in Eq. 7 is the "general" case, where $x$ and $z^{(1)}, \dots, z^{(S)}$ are not related by any structured function $h$. On the other hand, Eq. 9 describes our case, where $h$ satisfies Eq. 8. We only obtain Eq. 9 under special assumptions (think: circles have more special properties than squares).
>
> *$p_\theta(x)$ admits multiple arguments.* Indeed, this is the case. Here, $p_\theta(a)$ just means "the likelihood of the random variable $a$ given the model parameterized by $\theta$." This overloading of $p_\theta$ is common practice. For example, in DDPM (Ho et. al, 2020), $p_\theta$ takes just $x$ (the modeled likelihood), but is also overloaded to take the joint distribution $x_0, \dots, x_T$ (see the first sentence of Section 2) and a single conditional variable $x_{t-1} | x_t$ (Eq. 1). You can see similar overloading for $q$ (throughout Section 2). Ultimately, they are different densities over different variables related by $h$ that happen to coincide when $h$ is a volume-preserving map, which is why Eq. 9 is so useful.
>
> *Understanding Eq. 15.* Note that we are considering the log probabilities in Eq. 15. The sum of log probabilities is equal to the product of the probabilities. Therefore we equivalently have $p(x) = p(z^{(1)}) p(z^{(2)}|x^{(1)}) \cdots p(z^{(S)} | z^{(1)}, \dots, z^{(S-1)})$. The RHS of this equality becomes the joint likelihood $p(z^{(1)}, \dots, z^{(S)})$ by the probabilistic chain rule. Therefore this is just an application of Eq. 9. Many likelihood models consider log probabilities rather than probabilities for numerical stability.
>
> *Volume preservation.* We note that the volume preservation property only holds when considering the entire map $h(x) = (z^{(1)}, \dots, z^{(S)})$. It is not clear what precisely this means for just the relationship between $z^{(s)}$ and $z^{(s - 1)}$. Intuitively, converting from $z^s$ and $z^{s-1}$ can be seen as a super-resolution step. Therefore, I do not see evidence that $p(z^{(s)} | z^{(<s)})$ should be equal across $s$, nor that an invertible transformation of the form $h(z^{(s-1)}) = z^{(s)}$ should exist. Super-resolution is an ill-posed task (which we really can only achieve by relying on a very strong prior -- in our case, the prior of natural images), so rigorous relations between different scales may be difficult. However, we are happy that the reviewer has put thought in this direction, and would be happy to discuss further.
>
> *Concrete derivation for $p(x)$.* Note again that the overloading of $p$ is quite common (see above answer). We do provide a derivation for Eq. 9 in the proof for Lemma 4.1 (Appendix A). Can the reviewer elaborate on why this derivation is not sufficient?
>
> **On claims.**
> Our paper does not claim to have an improved loss over other models. Our claim is precisely that we enable a tractable likelihood in hierarchical models, whereas other models do not. This provides many downstream applications related to likelihood modeling, including density estimation, out-of-distribution detection, and neural compression, which we evaluate our models on.
>
>
> Does this make sense? Please let us know if you have any other concerns.
>
> [1] NICE: Nonlinear Independent Components Estimation. https://arxiv.org/abs/1410.8516

---

> > ### Comment · Reviewer_ocEJ · 2023-11-21
> > **resp**
> >
> > I think I'm starting to see some light. So Lemma 4.1. says that we can evaluate equivalently the $p_x(x)$ by instead evaluating $p_z(h(x))$, whatever the $p_z$ is as long as $h$ is "nice"; and here i'm writing explicitly the two densities as operating over different random variables. Then the image-$p_\theta$ (left) in eq 9 can be evaluated by evaluating the scale-$p_\theta$ (right) in eq 9, which has different input space but results in same density values.
> >
> > Is this correct?
> >
> > Few additional questions spring to mind from this.
> >
> > 1. In the introduction the motivation of the work is that we want to only consider $p(z^S)$ and thus avoid the joint modelling $p(z^1, \ldots, z^S)$ since it necessitates expensive marginalising of lower scales. But eq 15 is a joint over all scales. Surely you then need to now integrate out all the $z^1, \ldots, z^{S-1}$ out from eq 15 to arrive at the desired likelihood $p(z^S)$ or $p(x)$ and integrate all the intermediate scale paths that can arrive at $x$. If you don't marginalise, then we still have multiple possible assignments $z^1,\ldots,z^S$ that can be sampled from eq 15 RHS. We could sample few times, and get different generations, or different density evaluations wrt the intermediate scales. Isn't this an issue?
> >
> > 2. In introduction you claim that the joint $p(z^1,\ldots, z^S)$ is somehow not good, and instead you want $p(z^S)$. But surely we know all the scales of the data exactly, so why not want to use this information? Wouldn't it make the model much better if we compute the scales of data, and then use multiple likelihoods? Intuitively you are then decomposing the "full" likelihood into richer "sub-likelihoods" that represent how likely are different representation of the datapoint. Can you comment on this?
> >
> > 3. The volume-preserving map needs to have a unit log-determinant. This is very limited, and I would assume that the map can only "rotate" the data in the sense that it can't really modify it in any ways. Even affine transforms are not allowed, which is the simplest function that one usually thinks of. It is quite difficult to even invent a simple example volume-preserving map. Can you give an example? Is this intuition correct, and what is the capacity of volume-preserving maps? Can it destroy information or add information? Does it need to preserve information? The Laplacian pyramid for example destroys information unless you construct the full stack. In fig 2 we see only 3 scales: surely this is not sufficient to represent all the dog information? Finally, you claim that $||h(x)||_2 = ||x||_2$. Given that $x$ is an image of size CWH, and $h(x)$ is an image of size $SCWH$, doesn't this pose problems? How is the _2 norm defined? I also think that you include the original image as $z^S = x$, which means that this equality can't be true if you have $x$ on both sides. Can you comment?
> >
> > 4. What is ultimately the difference between your model and earlier cascade models (eg. Ho et al)? Both models seem to have quite similar model definitions, and eq 15 looks pretty "standard". Is the only difference the choice of the scaling operator, or is there also some likelihood evaluation differences?
> >
> > 5. What is "likelihood training and evaluation capabilities"? Earlier super-resolution diffusion models clearly are able to do generative modelling and density estimation. They clearly have ELBOs that are likelihood estimates. What extra you can do that others cant do?

---

> > > ### Author Response · Authors · 2023-11-21
> > >
> > > Yes, indeed, that is precisely the case. To answer your further questions:
> > >
> > > 1.  We do not need to integrate out the latent variables because we use Lemma 4.1 (i.e., Eq. 9). Since $h$ is invertible, the statement that there are multiple possible assignments of $z^{(1)}, \dots, z^{(S)}$ for any $x$ is not true. So there is only one possible assignment of $z^{(1)}, \dots, z^{(S)}$, and the scales have the same likelihood under our model $p_\theta$ due to Lemma 4.1. So there are no issues of sampling different random variables or getting different density evaluations given a fixed $x$.
> > >
> > > 2.  Since we want to produce a valid likelihood model, we need to a way to evaluate the likelihood of the data $x$ given the model parameterized by $\theta$ --- i.e., $p_\theta(x)$. This cannot trivially be obtained from $p_\theta(z^{(1)}, \dots, z^{(S)})$. $p_\theta(x)$ is of particular importance because many downstream tasks such as density estimation, out-of-distribution detection, and neural compression specifically involve this quantity (and not any other quantity such as  $p_\theta(z^{(1)}, \dots, z^{(S)})$). While the information from the scales is important, it is a different consideration than tractable likelihood evaluation.
> > >
> > > 3.  Indeed, in the linear case where $\mathcal{X}$ and $\mathcal{Z} = (\mathcal{Z}^{(1)} \times \cdots \times \mathcal{Z}^{(S)})$ are equidimensional, the volume-preservation property reduces to a rotation. However, this is not restrictive, since the important consideration in hierarchical models is simply a representation that explicitly decomposes a signal into a sequence of coarse-to-fine scales. As discussed, the wavelet transform and Laplacian pyramid decomposition are two such maps. Ultimately, volume-preserving maps cannot add or destroy information, since the map must be invertible. Therefore, the scales in Figure 2 are actually sufficient to reconstruct the entire image. Given an image of size $CWH$, $h(x)$ is not of size $SCWH$. Rather it is a sequence of images of size $CWH, C(W/2)(H/2), C(W/4)(H/4)$, etc. Finally, we do not make the claim that $||h(x)|| = ||x||$, this generally understood (orthonormality in wavelets and tight frames in Laplacian pyramids -- see Unser et. al, 2011). The $||\cdot||_2$ norm is the Frobenius norm.
> > >
> > > 4. Yes, we obtain significantly improved likelihood evaluation performance due to the choice of $h$, which is different from Ho et. al, 2022. Please see Appendix C and Tables 4-6 for a comparison against the scheme in Ho et. al, 2022.
> > >
> > > 5. Likelihood training and evaluation capabilities relies on the ability to evaluate the quantity $p_\theta(x)$. To the best of our knowledge, we have not found any earlier hierarchical diffusion models that can do this. Therefore, no earlier hierarchical diffusion models are capable of density estimation, which, recall, is obtaining $p_\theta(x) \approx p(x)$, where $p$ is the true data density. If the reviewer is aware of any such examples, we are happy to take a look at them. We do not claim to be the first hierarchical diffusion model to perform generative modeling -- there are many such models including Ho et. al.

---

> > > > ### Comment · Reviewer_ocEJ · 2023-11-21
> > > > **resp**
> > > >
> > > > Thanks for your patience. My concerns are addressed, and I think this is a substantial contribution to the field, and would be happy to see this accepted.

---

### Official Review · Reviewer_PAyK · 2023-11-08

**Soundness:** 2 fair
**Presentation:** 2 fair
**Contribution:** 2 fair
**Rating:** 5
**Confidence:** 4

**Summary:**

This paper proposes the cascaded diffusion models for high-fidelity image synthesis that also works as a likelihood model, by overcoming the intractability issue of the likelihood function in multi-scale diffusion models. To do this, this paper provides some hierarchical model of the latent variables, to reduce the marginal likelihood problem into the cascaded conditional likelihood problem. To do this, a class of transformations called hierarchical volume-preserving maps. Some special yet natural instances of the hierarchical volume-preserving maps include Laplacian pyramids, where the image data is divided into its low-pass filter part and its counterparts sequentially, and the wavelet decomposition, which is more general and more rank-preserving transformation, first orthogonalize the image into four filters (low-low, low-high, high-low, high-high) which is also called subbands. Then, the subbands that include high-pass filter in at least one coordinate (vertical or horizontal) is kept and low-low subband is hierarchically downsampled and again taken the wavelet transform.

Then, the likelihood training of this cascaded diffusion model is done by the factorization of the variational objective of the diffusion model into the sum of the conditional variational objective with respect to each hierarchical latent variables. Finally, the paper suggests the connection of this paper to the optimal transport concept, by providing that the Wasserstein-p distance between the approximate and exact score is upper-bounded by the cascaded diffusion objective. The experiments show that the cascaded volume-preserving diffusion model yields better likelihood measure compared to other existing flow-based models and diffusion models. Furthermore,

**Strengths:**

* The writing of the concepts is clear and easy to understand, starting from the necessity of the cascaded latent variable models, and introduce the diffusion models that are involved with this LVM with the volume-preserving map. And in case of the volume-preserving maps, the paper showed that like in the existing original diffusion models, this hierarchical volume-preserving diffusion models also works  as the minimizer of the Wasserstein distance.
 * This paper showed superior performance in case of the log-likelihood measures of CIFAR-10 datasets, compared to the existing semi-autoregressive methods.

**Weaknesses:**

* The use of semi-autoregressive hierarchical diffusion models is a common approach, including the use of wavelets, latent variables, and null-space vectors for upsampling the images. The use of cascaded volume-preserving maps do not seem to give differences to these existing works, and the related works on multi-scale diffusion models should be more carefully added.
 * The optimal transport theorem (5.1) does not enough evidence that the hierarchical volume-preserving map is more feasible to use for matching score function than the original diffusion models. Kwon et al. (2022), already showed the similar theorem such that the diffusion model upper bounds the Wasserstein distance between the approximate and the exact score.

D. Kwon et al, "Score-based Generative Modeling Secretly Minimizes the Wasserstein Distance" (2022)

**Questions:**

* According to the specific features of the hierarchical models, the architecture description in appendix B should be more precisely described. Now it is not easy to understand which architecture is used for training multiscale models (single model for all scales, or multiple models such that the (conditional) score for each scale is learned in each model?), including the number of parameters. (This might not be a problem since this paper used the same architecture to the VDM paper.)

===============

 * Many parentheses are not closed; I recommend using ( \left[, \right] ) and ( \left(, \right) ) command.
 * In the first paragraph of Section 5.1, $z^{(1)},\cdots,z^{(S)} = h(x)$ is considered to be the abuse of denoting the hierarchical volume-preserving map. Please consider using clearer notation to represent this.

**Details Of Ethics Concerns:**

None.

---

> ### Author Response · Authors · 2023-11-20
> **Reply to Reviewer PAyK**
>
> We are grateful for the reviewer's appreciation of the clarity and relevance of our work, and for raising insightful concerns. Below we respond to each comment.
>
> **Differences from existing hierarchical diffusion models.** Indeed, hierarchical diffusion models exist in the literature, and we thank the reviewer for the references, which we have added to the text. However, as discussed, these models do not support tractable likelihood computation. The main contribution of our work is to show the conditions under which hierarchical models such as these are also feasible likelihood models (Eq. 8), which enable many downstream analyses (anomaly detection, data compression, data enrichment, etc.). To further support the importance of our hierarchical volume-preserving property and the **exact** likelihood computation it enables (Eqs. 8-9), we provide additional experiments in Appendix C and Tables 4-6 that compare our model against $h$ that does not satisfy Eq. 8.
>
> **How is Theorem 5.1 different from Kwon et al., 2022?** While the theorems appear similar, Kwon et. al, 2022 differs significantly from our approach. Namely, Kwon et. al, 2022 (as well as several other works - De Bortoli et. al, 2021, Chen et. al, 2021, Lipman et. al, 2022, Shi et. al, 2022, all cited in our work) consider optimal transport in the data space $\mathbf{R}^d$, where $d$ is the image size, e.g., 32x32x3 = 3072 for CIFAR10, whereas we consider optimal transport on the 2D (i.e., discretized $\mathbf{R}^2$) spatial grid of image and score pixels. In particular, Theorem 5.1 relates multiscale score matching to score matching under the Earth Mover’s Distance ($p=1$), which on spatially structured data (e.g. images, audio) has many well-known properties and is regarded as ***a superior metric to the L2 distance used in original diffusion models*** (see Rubner et al., 2000). Ultimately, this results in distinctly different theoretical properties from the above-mentioned related work (including Kwon et. al, 2022), and is also the first such connection to EMD on images.
>
> **Architecture description in Appendix B.** Indeed, there is a separate score model for each scale. We have added a sentence clarifying this point in Appendix B. We have also added the number of parameters for each model (also Appendix B).
>
> Minor edits:
>
> **Closing parentheses.** Thank you for bringing this to our attention. Indeed, the \left( and \right) latex operations are very helpful, and we have incorporated them here and elsewhere in the text.
>
> **$h(x)$ notation.** We apologize for the confusion. The range of $h$ is the cartesian product of the $S$ scale spaces, i.e., $h: \mathcal{X} \rightarrow \mathcal{Z}^{(1)} \times \mathcal{Z}^{(2)} \times \dots \times \mathcal{Z}^{(S)}$. We have reversed the equality and added parentheses around the RHS in the equation $h(x) = (z^{(1)}, \dots, z^{(S)})$ to alleviate this confusion, and further clarified the functional definition of $h$ in Sections 3.1 and 5.1.

---

> ### Comment · Reviewer_PAyK · 2023-11-22
> **Reply to Response (1)**
>
> We greatly appreciate the detailed response of the authors. I now admit the consistency of the optimal transport theorem that connects the EMD loss to the score function with the rebuttal with Reviewer y69M. On the other hand, I am curious about the following points.
>
> 1. I have been concerned that the volume-preserving map with respect to the orthogonal wavelets is actually the scaling of the wavelet bases (which is equal to normalization) to preserve volume, and I doubt that the significant gain in terms of NLL (=BPD) is obtained by this aspect of loss design. In a practical point of view, the scaling in each wavelet basis leads to the scaling in the forward (thus reverse process) of the conditional diffusion models with downsampled wavelet coefficients, and in my opinion, the FID gain (and maybe the BPD gain) comes from this rescaling of the diffusion process.
>
> 2. As I have concerned, the (conditional) likelihood-based diffusion models that the paper used utilizes multiple neural netowrks, using one network for different scale to work as hierarchical models. The final version of the manuscript should also contain the input-output relationship and the architecture of the score model (including the number of Resblocks, the Attention resolutions, so on....) I confirmed that the number of parameters are considered in Appendix B, but the wall-clock time (or the FLOPs, if possible) is not considered yet. This makes me suspect that the gain of this paper is because of arbitrarily bulked models.
>
> For these unresolved issues, I keep my current review score.
>
> * * *
>
> __Clarity of paper__
>
> (Overall) Please do not mix up $\texttt{\citet}$ (without parenthesis) and $\texttt{\citep}$ (with parenthesis). \
> (Page 1) To regain $p_\theta(x)$ evaluation capabilities $\to$ For $p_\theta(x)$ to regain evaluation capabilities \
> (Overall) Please use $\texttt{\paragraph}$ instead of bold text for the title of paragraphs.

---

> ### Author Response · Authors · 2023-11-23
> **Reply to Reviewer PAyK (1/2)**
>
> We thank the reviewer for their continued and thorough efforts in evaluating our work, and for their additional suggestions and questions. Below we provide our responses:
>
> **Further concerns.**
>
> 1.  **Scaling of the diffusion process.** Indeed, any scaling of the coefficients results in a scaling of the respective diffusion processes. However, we maintain that this rescaling would not change the model performance in terms of FID or BPD. In the former, the FID is simply a function of embeddings of the sampled images, so it is agnostic to the sampling process (e.g., agnostic to whether images were sampled autoregressively or all at once, or via scaled or unscaled diffusion processes). In the latter, any alternative scaling of the wavelet transform is no longer volume-preserving, and incurs a penalty in the form of the $\sqrt{\text{det}\left(\left[\frac{\partial}{\partial x} h(x)\right]^T\left[\frac{\partial}{\partial x} h(x)\right]\right)}$ term in the change-of-variables formula for $p_\theta(x)$ (Eq. 20). In terms of the *negative log*-probability (proportional to BPD), this results in a correction factor that would actually cancel out the change in log probability due to scaling coefficients. We do not consider these non-volume-preserving cases because this correction term can be difficult to compute or derive.
>
>     However, we do understand the reviewer's concern. We interpret the underlying question as **whether the wavelet transform (and Laplacian pyramid) results in a "real" improvement in model performance over vanilla diffusion models, or just a "rescaling" of the loss in some trivial manner**. We firmly believe that it is the former, and elucidate this in the following experiment. Taking the learned conditional score networks $s_\theta(z^{(s)}|z^{(<s)}, t)$ in a trained W-PCDM model on CIFAR10, we observe that we can reconstruct a conditional "vanilla" score $s_\theta(x, t)$ by applying $h^{-1}$ to the hierarchical conditional scores -- i.e., the inverse of Eqs. 31-32. This holds due to the simple form of the score (i.e., $\frac{x + \epsilon}{\sigma}$, Vincent, 2011) when q(x_k|x_0) is Gaussian (which is our case, see e.g., Ho et. al, 2020). After this transformation, we evaluate the ELBO of this "vanilla" diffusion model on the CIFAR10 test set. We ultimately obtain the same gain in performance as the W-PCDM (i.e. $2.65 \rightarrow 2.35 BPD$), even though the present formulation no longer transforms the pixel scores in any way after they are computed (only to convert from the wavelet scores to the pixel scores). This tells us that the gain modeling performance is unrelated to the scaling of the wavelet coefficients.
>
> 2. **Architecture of the score model.** Please note that we do discuss in-depth the architecture of the score model in Appendix B -- in summary, we use the VDM architecture with no changes except for the input image, which now concatenates the noise-perturbed scale variable $\tilde{z}^{(s)}$ with the conditional information $z^{(<s)}$. We have also added an explicit description of the input-output relationship of our model in Appendix B.
>
>     **FLOPs.** While our models have more total parameters, our W-PCDM implementation actually has significantly fewer FLOPs than the standard diffusion model, and is therefore both faster and smaller in memory footprint. This is mainly due to the smaller input sizes at each scale. Overall, we approximate that the sum total FLOPs across all scales (versus a vanilla VDM model) is reduced by at least a factor of 2. (For more details, see "Calculating FLOPs of W-PCDM.") As discussed in [1], FLOPs and throughput are as important indicators for model efficiency / complexity as the number of parameters.
>
>     **Wall-clock time.** On an NVIDIA A6000 GPU, we observe that the reduction in wall-clock time between W-PCDM and a standard VDM follows closely with our approximations. In our CIFAR10 model, there is a 2.3x reduction in wall-clock time during inference versus VDM (26 min to evaluate 30K examples versus 1 hour).
>
> [1] The Efficiency Misnomer. https://arxiv.org/pdf/2110.12894.pdf

---

> ### Author Response · Authors · 2023-11-23
> **Reply to Reviewer PAyK (2/2)**
>
> **Calculating FLOPs of W-PCDM.** Note that for an input image of size $C \times H \times W$ (where $C$, $H$, and $W$ are the channels, width, and height respectively) the largest input resolution of any scale in the W-PCDM model is size $3C \times (H / 2) \times (W / 2)$. Each subsequent model then takes an input of size $3C \times (H / 2^s) \times (W / 2^s)$ for $s = 1, \dots S-1$, with the final scale being $C \times (H / 2^S) \times (W / 2^S)$. A U-Net is comprised almost entirely of convolutional layers and attention layers -- plus relatively inexpensive element-wise and group-wise nonlinearities. The FLOPs of a convolutional layer is usually $C_{in} \times C_{out} \times (H_{kernel} \times W_{kernel}) \times (H_{out} \times W_{out})$. Therefore, fixing the network architecture, the FLOPS of a convolutional layer of the W-PCDM model at the first scale level is approximately $1/4$ that of the original model on the original image ($H_{out}$ and $W_{out}$ are halved w.r.t. the original model). Some exceptions are the first and last convolutional layers of the model, and upsampling/downsampling layers. The FLOPs of an attention layer is quadratic in the image size, i.e., $(H \times W) \times (H \times W)$, and thus reduces by 1/16 in the first scale of a wavelet model. Applying the same argument to the remaining scales, one can find that the FLOPs at each scale is approximately $2^{-2s}$ that of the original FLOPs, where $s$ is the scale level. The sum of all FLOPs across scales is thus less than $1/2$ that of the original FLOPs (by applying a crude geometric bound $1/4 \sum_{s=0}^S (1 / 2)^s \leq 1 / 2$).
>
> **Clarity of paper.**
>
> *Mixing up $\texttt{\citet}$ and $\texttt{\citep}$.* Thank you for this catch. We have standardized all citations with $\texttt{\citep}$.
>
> *Rephrasing page 1 "To regain $p_\theta(x)$ evaluation capabilities", and using $\texttt{\paragraph}$ for paragraph titles.* Again, thank you kindly for the suggestions -- we have incorporated them.

---

### Author Response · Authors · 2023-11-20
**General response to all reviewers**

We are very grateful to all reviewers for their positive comments and helpful feedback. All reviewers found the paper generally **clear and well-written**, with **impressive empirical results**. Reviewers y69M and ocEj note the importance of the problem tackled by our work, while reviewers PAyK, hzLh, and ocEj found the connection between hierarchical modeling and optimal transport particularly interesting.

We have taken all comments and suggestions into account and believe they have substantially improved the paper. We have made several changes including **additional citations**, **clarified exposition**, **comprehensive proofs (Appendix A)**, and **ablations demonstrating the importance of our proposed hierarchical volume-preservation property (Eq. 8) (Appendix C)**. Updated text is colored in blue. We address reviewer’s comments on a point-by-point basis in the individual reviews.

---

### Public Comment · ~Yirong_Shen2 · 2025-08-19

This is a strong and insightful paper with clear theoretical contributions and convincing experiments. I did notice a minor issue in Table 1: the reference for *Improved DDPM* should be

Nichol, Alexander Quinn, and Prafulla Dhariwal. Improved denoising diffusion probabilistic models. ICML, 2021.

rather than

Prafulla Dhariwal and Alexander Nichol. Diffusion models beat GANs on image synthesis. NeurIPS, 2021.

---

### Meta-Review · Area_Chair_pX1R · 2023-12-03

**Metareview:**

The authors consider cascaded models (multi-scale generative models geared towards high-resolution generation). Existing such models suffer from intractability of computation of the likelihood function (because of extraneous variables at intermediate scales); tractable likelihood function allows downstream applications.

In this paper, they solve this problem, giving the first multiscale diffusion models with tractable likelihood by using hierarchical volume-preserving maps. Natural well-studied models based on Laplacian/wavelet decomposition fit into their framework. They improve SOTA on benchmarks including density estimation, lossless compression, and out-of-distribution detection. They also give connections to optimal transport.

The strength of the paper comes from it being the first result of its type, theoretical grounding and convincing experiments.

**Justification For Why Not Higher Score:**

The bar for oral is high, reserved for papers with greater significance beyond the immediate topic.

**Justification For Why Not Lower Score:**

The result is the first of its type on an important problem, based on principled math and with convincing experiments. 3/4 reviewers gave strong accept.

---

### Decision · Program_Chairs · 2024-01-16

Accept (spotlight)